# VeriCoT: Neuro-Symbolic Chain-of-Thought Validation via Logical Consistency Checks

**Yu Feng**[†*], **Nathaniel Weir**♠, **Kaj Bostrom**♠, **Sam Bayless**♠,
**Darion Cassel**♠, **Sapana Chaudhary**♠, **Benjamin Kiesl-Reiter**♠, **Huzefa Rangwala**♠
[†]University of Pennsylvania   ♠Amazon Web Services

## Abstract

LLMs can perform multi-step reasoning through Chain-of-Thought (CoT), but they cannot reliably verify their own logic. Even when they reach correct answers, the underlying reasoning may be flawed, undermining trust in high-stakes scenarios. To mitigate this issue, we introduce VeriCoT, a neuro-symbolic method that extracts and verifies formal logical arguments from CoT reasoning. VeriCoT formalizes each CoT reasoning step into first-order logic and identifies premises that ground the argument in source context, commonsense knowledge, or prior reasoning steps. The symbolic representation enables automated solvers to verify logical validity while the NL premises allow humans and systems to identify ungrounded or fallacious reasoning steps. Experiments on the ProofWriter, LegalBench-SARA, and BioASQ datasets show VeriCoT effectively identifies flawed reasoning, and serves as a strong predictor of final answer correctness. We also leverage VeriCoT's verification signal for (1) inference-time self-reflection, (2) supervised fine-tuning (SFT) on VeriCoT-distilled datasets and (3) preference fine-tuning (PFT) with direct preference optimization (DPO) using verification-based pairwise rewards, further improving reasoning validity and accuracy.

## 1 Introduction

Chain-of-Thought (CoT) (Wei et al., 2022) in natural language (NL) has emerged as a powerful technique for improving large language model (LLM) performance on tasks that require reasoning. LLMs like DeepSeek-R1 (DeepSeek AI, 2025) and OpenAI's o1 (OpenAI, 2024b) demonstrate strong reasoning capabilities using CoT. However, these models still frequently make logical errors in their reasoning chains, even when the final answer is correct (OpenAI, 2024a). For example, an LLM might generate the question answering (QA) CoT depicted in Figure 1, which leads to a correct answer, but an intermediate step might be false (e.g., *"Charlie is at most 15"* instead of *"at most 18"*). When LLMs generate logically invalid CoT steps, it hinders their trust and usefulness, regardless of whether the final answers are correct. This issue is particularly critical in high-stakes domains such as biomedical or legal reasoning, where users consider the accuracy of the reasoning path as important as that of the final answer. This behavior can be attributed in part to an inherent limitation of LMs: they predict text without an explicit mechanism for verifying the logical validity of the resulting semantics (Bender & Koller, 2020; Ji et al., 2023)

Prior work has attempted to mitigate this through self-refinement of the model's output based on diverse feedback sources, such as dynamic retrieval of facts from external knowledge bases (Peng et al., 2023), a separate critic model (Paul et al., 2024), dynamic use of program execution (Chen et al., 2024; Olausson et al., 2024), or symbolic checking (Olausson et al., 2023; Ye et al., 2023; Ling et al., 2023; Pan et al., 2023; Quan et al., 2024a; Lalwani et al., 2025). However, these approaches do not ensure the logical validity of the *entirety* of an LLM's output. The recently introduced Explanation-Refiner (Quan et al., 2024b) makes progress towards this goal by performing iterative autoformalization and refinement of NL explanations elicited from an LLM, guided by a theorem prover, for NLI task. However, there is yet to be a verification method that simultaneously (1) applies over LLM Chain-of-Thought steps, (2) formalizes the context grounding of every reasoning step, and (3) improves LLM reasoner capabilities with respect to logical validity in domains other than code/math. To address these limitations, we introduce VeriCoT: Symbolic **Veri**fication for

---

*Work done during internship at AWS

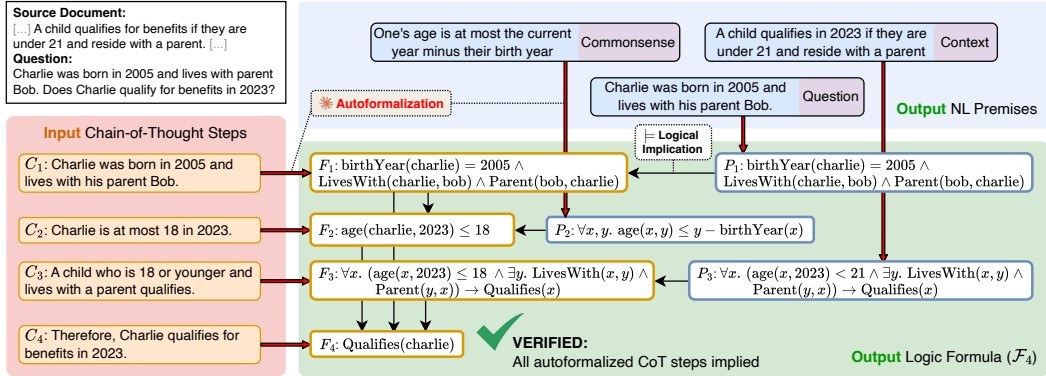

Figure 1: VERICOT verification of a Chain-of-Thought for the SARA dataset (Holzenberger et al., 2020). Even if the final answer is correct, a CoT that contains an invalid step hurts user trust and raises questions of LLM faithfulness. As shown in §2.1, VERICOT autoformalizes each step of the CoT into symbolic logic, producing a formula that ensures each one follows logically from a distilled list of NL premises, each of which it annotates with their source type (e.g. Commonsense or Context) If the CoT cannot be represented this way, it is unverifiable.

**Chain-of-Thought**, a neuro-symbolic algorithm that grounds and validates the logical consistency of Chain-of-Thought reasoning, identifying logical errors and making implicit premises explicit.

VERICOT maintains a growing set of first-order-logic (FOL) premises inferred from NL context. It autoformalizes each CoT step into a first-order-logic formula, and uses a constraint solver to check whether it is logically entailed by the premises and previously formalized steps. If the step isn't entailed, VERICOT either autoformalizes a sufficient set of supporting premises from available context (question text, documents, or common sense) or identifies a reason why the step cannot be validated (e.g., because a premise could not be inferred from context, or because the step contradicts the inferred premises). Through this process, VERICOT provides multi-faceted feedback. It identifies **whether** a CoT can be represented in formal logic, **how** the CoT's steps are logically supported, and **what** underlying NL premises need to be accepted in order to accept the CoT's reasoning. To our knowledge, this is the first neuro-symbolic validator of CoT traces in non-math/code domains.

Our evaluation demonstrates that VERICOT can detect ungrounded or incorrect reasoning, and VERICOT-validation is a strong predictor of final answer correctness, with validated CoT attaining higher precision than task-level accuracy (§3.3). Building on this verification capability, we leverage VERICOT to actively enhance an LLM's reasoning (§3.4). First, we use VERICOT for inference-time self-reflection, where its validity-oriented feedback prompts the model to self-correct, yielding an average 46% relative improvement in CoT verification pass rate and consistent 41% relative gains in producing accurate and verifiable task outcomes across multiple datasets. Second, beyond inference-time correction, we build upon recent work using formal reasoning for RL verification (Leang et al., 2025) and use VERICOT to create a high-fidelity dataset of verified CoTs for Supervised Fine-tuning (SFT) and as a source of pairwise reward signals for preference fine-tuning with direct preference optimization (DPO) (Rafailov et al., 2024). These fine-tuning strategies improve the model's ability to generate logically consistent CoTs by 18% (relative) while matching or exceeding base task accuracy in less formal domains like biomedical and legal reasoning. We distill CoTs from a substantially stronger teacher model and retain only those that pass VERICOT's checks, yielding higher precision supervision than randomly sampled unfiltered traces when gold task labels are unavailable.

## 2 NEURO-SYMBOLIC CoT VERIFICATION ALGORITHM

Our goal is to validate the correctness of Chain-of-Thought reasoning by ensuring that individual reasoning steps can be translated into consistent first-order-logic formulas. We assume we are given a natural language *context* (which includes a question, an optional conversation history, and a source document with information relevant to the question) and a *CoT* consisting of NL steps $C_1, \ldots, C_n$.

As described in Alg. 1, VERICOT autoformalizes each step $C_i$ into a first-order logic formula $F_i$, then uses a constraint solver to determine whether $F_i$ can be logically derived from our current

---

**Algorithm 1:** VERICOT Overview

---

1. **Initialize:** Set $\mathcal{F}_0 = \emptyset$, $\mathcal{P}_0 = \emptyset$, $errors = \emptyset$.

2. **For each CoT step $C_i$ ($i = 1, \ldots, n$):**

   (a) **Autoformalization (§2.2):** Formalize step $C_i$ into logic formula $F_i$. If $C_i$ cannot be adequately expressed in our supported logic fragment or syntax errors persisted after multiple attempts:
       $errors$.add($\langle i, \textbf{untranslatable} \rangle$). Continue on to next CoT step $C_{i+1}$.

   (b) **Consistency check:** If the formalized $F_i$ contradicts our established knowledge ($\mathcal{F}_{i-1} \models \neg F_i$):
       $errors$.add($\langle i, \textbf{contradiction} \rangle$). Continue on to next CoT step $C_{i+1}$.

   (c) **Entailment check:** If $F_i$ is logically implied by our established knowledge ($\mathcal{F}_{i-1} \models F_i$):
       set $\mathcal{F}_i = \mathcal{F}_{i-1} \cup \{F_i\}$ and $\mathcal{P}_i = \mathcal{P}_{i-1}$ and continue on to next step $C_{i+1}$.
       Otherwise, try to generate supporting premises:

   (d) **Premise generation (§2.3):** Generate premise $P_i$ to support $F_i$.[1] Check if $P_i$ is consistent with established statements ($\mathcal{F}_{i-1} \not\models \neg P_i$)
       *Optional*: Use **LLM-as-judge evaluation (§2.4)** to check that $P_i$ is attributable to source context.
       If $P_i$ still fails to helps entail $F_i$ ($\mathcal{F}_{i-1} \cup \{P_i\} \not\models F_i$): $errors$.add($\langle i, \textbf{ungrounded} \rangle$).
       set $\mathcal{P}_i = \mathcal{P}_{i-1} \cup \{P_i\}$ and $\mathcal{F}_i = \mathcal{F}_{i-1} \cup \{P_i\} \cup \{F_i\}$ and continue on to next CoT step $C_{i+1}$.

3. **Return:** Sets $\mathcal{P}_n$, $\mathcal{F}_n$, $errors$.

---

knowledge ($\mathcal{F}_{i-1}$) or requires additional premises. We distinguish the following logical relationships between a new statement and our established knowledge:

- $F_i$ is *entailed* by $\mathcal{F}_{i-1}$, meaning it necessarily follows from what we already know ($\mathcal{F}_{i-1} \models F_i$),
- $F_i$ is *contradicted* by $\mathcal{F}_{i-1}$, meaning it is inconsistent with what we know ($\mathcal{F}_{i-1} \models \neg F_i$),
- $F_i$ is *consistent* with, but not entailed by, what we already know ($\mathcal{F}_{i-1} \not\models F_i$, $\mathcal{F}_{i-1} \not\models \neg F_i$).

VERICOT identifies a CoT as *valid* if it can infer a consistent set of logical premises from the NL context that are sufficient to entail every step of the CoT. If valid, VERICOT returns:

1. A self-consistent set of FOL premises $\mathcal{P}$, where each premise $P \in \mathcal{P}$ represents the formalization of an NL source (either from the provided context, or from common sense).
2. For each NL step $C_i$, a FOL formula $F_i$ (the formalization of $C_i$) such that $\mathcal{P} \models F_i$.

If invalid, VERICOT identifies error reasons for all steps that were not entailed. These reasons are used as a feedback signal for inference-time self-reflection or fine-tuning data distillation (§3.4):

1. **Ungrounded**: VERICOT could not identify a sufficient, non-contradictory set of strengthening premises from context that would entail $F_i$ (e.g., $\mathcal{F}_{i-1} \cup \{P_i\} \not\models F_i$).
2. **Contradiction**: Existing statements contradict $F_i$ (e.g., $\mathcal{F}_{i-1} \models \neg F_i$).
3. **Untranslatable**: $C_i$ could not be translated into the FOL subset supported by VERICOT, or syntax errors persisted after multiple attempts.

Logic formulas are encoded in `SMT-LIB` (Barrett et al., 2016), which supports a fragment of first-order logic with theories including linear arithmetic, uninterpreted functions, and quantifiers. We use the SMT solver Z3 (de Moura & Bjørner, 2008) to perform the logical consistency and entailment checks described above. To build intuition, we first walk through a simplified example that demonstrates our approach on the CoT depicted in Figure 1.

## 2.1 EXAMPLE OF COT VERIFICATION

**Step 1: Infer premises from the user's question.** For the first step, "*Charlie was born in 2005 and lives with his parent Bob,*" VERICOT autoformalizes it as follows (see §2.2):

$$F_1 := birthYear(charlie) = 2005 \land LivesWith(charlie, bob) \land Parent(bob, charlie)$$

---

[1] $P_i$ can be a conjunction of multiple statements that are then evaluated individually by LLM-as-judge (§2.4).

At this initial point, the set $\mathcal{F}_0$ of previously established statements is empty, so $\mathcal{F}_0 \nvDash F_1$. Since the current step's statement cannot be derived from $\mathcal{F}_0$, we check if supporting premises can be inferred from the provided context. Using an LLM, VERICOT is able to derive a supporting premise $P_1$ from the question (in this case, $P_1$ happens to be identical to $F_1$). We add $P_1$ to our premise set $\mathcal{P}_1$, confirm (via constraint solver) that $\mathcal{P}_1 \wedge \mathcal{F}_0 \vDash F_1$, and finally, add both (identical) statements to $\mathcal{F}_1$.

**Step 2: Infer premises from common sense.** For the second step, "*Charlie is at most 18 years old in 2023*," VERICOT autoformalizes it as:

$$F_2 := age(charlie, 2023) \leq 18$$

This statement cannot be derived from $\mathcal{F}_1$, which contains only information about Charlie's birth year and living situation, but no direct facts about his age. To bridge this gap, our premise generation approach (see §2.3) identifies a commonsense assertion relating age to birth year: "*Someone's age is at most the current year minus their birth year*", autoformalized as:

$$P_2 := \forall x, y.\, age(x, y) \leq y - birthYear(x)$$

We can derive $age(charlie, 2023) \leq 18$ from the combination of $\mathcal{F}_1$ and this new premise, so we obtain $\mathcal{P}_2$ by adding $P_2$ to $\mathcal{P}_1$, and we set $\mathcal{F}_2 = \mathcal{F}_1 \cup \{P_2, F_2\}$. Note that we reused the existing constant $charlie$ and the predicate $birthYear$ from our established vocabulary, introducing only the new predicate $age$ as needed. Also note that if the step had made an incorrect claim like "*Charlie is at most 15*," our verification would report the step as inconsistent because $P_2$ contradicts it.

**Step 3: Infer premises from source document.** For the third step, "*A child who is 18 or younger and lives with a parent qualifies*," we formalize it as:

$$F_3 := \forall x.\, (age(x, 2023) \leq 18 \wedge \exists y.\, LivesWith(x, y) \wedge Parent(y, x)) \rightarrow Qualifies(x)$$

This universal statement cannot be derived from $\mathcal{F}_2$. However, the source document contains a stronger statement, which VERICOT autoformalizes as a new premise, "*A child qualifies for benefits if they are under 21 and live with their parent*", we formalize it as:

$$P_3 := \forall x.\, (age(x, 2023) < 21 \wedge \exists y.\, LivesWith(x, y) \wedge Parent(y, x)) \rightarrow Qualifies(x)$$

With $\mathcal{P}_3$ strengthened by adding $P_3$ to $\mathcal{P}_2$, we can derive $\mathcal{P}_3 \wedge \mathcal{F}_2 \models F_2$, and so we update $\mathcal{F}_3 = \mathcal{F}_2 \cup \{P_3, F_3\}$ accordingly. Note that if this rule were not present in the source document or context, our verification would report the CoT step as ungrounded.

**Step 4: Conclusion.** The last step, "*Therefore, Charlie qualifies for benefits in 2023*," formalizes as:

$$F_4 := Qualifies(charlie)$$

This conclusion can now be derived from $\mathcal{F}_3$ without requiring additional premises: we know Charlie is at most 18 (hence under 21), lives with his parent Bob, and the qualification rule applies to all children meeting these criteria. Had the step wrongly said "*Charlie does not qualify for benefits in 2023*", we would have identified it as inconsistent with the reasoning trace.

Below we describe each modular component of VERICOT in more detail.

## 2.2 AUTOFORMALIZATION

Our autoformalization approach works in two stages, both using LLMs to translate NL into `SMT-LIB`. When attempting to translate a CoT step $C_i$ into its logical representation $F_i$, the first stage uses an LLM prompt that includes the previously produced logical vocabulary as context. The LLM is prompted to generate a structured, intermediate representation that combines `SMT-LIB` with metadata about which text in $C_i$ corresponds to which parts of the resulting $F_i$, using only the variables and types already present in the vocabulary.

The second stage extends the vocabulary if there are segments of the input text that the LLM deems relevant to the CoT's logical argument but cannot express with the existing vocabulary. In this stage, we use another LLM prompt to generate new `SMT-LIB` declarations (e.g., `declare-fun`, `declare-sort`). The new declarations are added to the vocabulary, and the first stage is attempted again. In our implementation, we allow this to repeat up to three times before giving up (in which case the current step is marked as untranslatable, or the current premise is discarded).

In the following example, when VERICOT initially attempts to translate $C_2$ ("Charlie is at most 18 years old in 2023") based on the available vocabulary, it first reports that the vocabulary is insufficient, then extends it accordingly to translate $C_2$ into $F_2$:

---

**Context:** Charlie was born in 2005 and lives with Bob.
**Document:** A child qualifies for benefits if they are under 21 and live with their parent.
**Question:** Does Charlie qualify for benefits in 2023?

**CoT Steps:**
$C_1$: "Charlie was born in 2005 and lives with parent Bob"
$C_2$: "Charlie is at most 18 years old in 2023"

**Previously Produced Vocabulary of $\mathcal{F}_1$:**

```
; represents a person
(declare-sort Person)
; specific person Charlie
(declare-const charlie Person)
; specific person Bob
(declare-const bob Person)
; birth year of a person
(declare-fun birth_year (Person) Int)
; whether person lives with another person
(declare-fun lives_with (Person Person) Bool)
```

**Initial (Failing) LLM Translation Output $F_2$:**

```
; current year 2023 -- UNTRANSLATABLE
(assert false)
; charlie age <18 -- UNTRANSLATABLE
(assert false)
```

**New Declarations Added to Vocabulary:**

```
; current year for calculation
(declare-const current_year Int)
; age of a person in a given year
(declare-fun age_in_year (Person Int) Int)
```

**Successful LLM Translation Output $F_2$:**

```
; current year 2023
(= current_year 2023)
; charlie age <18
(<= (age_in_year charlie current_year) 18)
```

---

Successful autoformalization allows us to represent the step as $F_2$, but not necessarily ensure that it is entailed by the growing logical formula, which is the focus of §2.3.

## 2.3 PREMISE GENERATION

Chain-of-Thought steps aren't always directly implied by existing statements. Instead, they might rely on information from the context, like problem details ("Charlie is Bob's son") or support documents ("a child qualifies for benefits if they are under 21"), or from common sense ("a father is a parent"). When a CoT step's formula $F_i$ is neither entailed nor contradicted by existing statements, VERICOT prompts an LLM to identify supporting premises from the context or commonsense.

We attempt to build a premise formula $P_i$ that is sufficient to entail $F_i$ as follows: We first generate multiple noncontradictory candidate NL premises and translate each of them into a logic formula using the autoformalization process from §2.2. We perform a round of premise regeneration if the natural language meaning of any new declarations from this step is not captured in the generated premises. For each candidate premise $p$, we then check whether it is consistent with the established statements (i.e., whether $\mathcal{F}_{i-1} \wedge p$ is satisfiable) and keep only those that are. Finally, we conjoin all remaining candidate premises to form the final premise formula $P_i$.

## 2.4 LLM-AS-JUDGE PREMISE EVALUATION

VERICOT ensures that the CoT is FOL-representable and presents a list of inferred premises as one necessary basis for accepting the CoT's logic. However, it has not verified that whether the premises themselves might be accepted. It may have enumerated the premise "the sky is purple" as necessary for accepting the CoT, but this is generally not an acceptable statement on its own. To provide greater assurance that the inferred premises are reliable, VERICOT uses an LLM-as-Judge approach (LLMaJ) to identify spurious premises after the premise generation process is complete. While the LLM generating premises in Alg. 1 can produce confabulations or omit relevant details, using LLMaJ reduces the likelihood that these errors go undetected.

We evaluate all premises using the LLMaJ approach. For premises inferred from source text, we provide judge LLMs with both the source document and the NL version of the premise, prompting them to evaluate whether the premise is attributable to the source text. For premises inferred from common sense, we prompt judge LLMs to evaluate whether the premise is acceptable given the provided context and targeted reasoning step (omitting the attribution requirement).

## 3 EXPERIMENTS

**Evaluation Models.** We use Claude-3.5-Sonnet-V2 (Anthropic, 2024) through API calls as the executor of the proposed VERICOT. We fine-tune Qwen2.5-7b-Instruct (Qwen Team, 2024).

| Method | Pass Rate | Precision | VCAR | Task Acc |
|---|---|---|---|---|
| **ProofWriter** | | | | |
| ER | 14.8 | 83.3 | 12.3 | 75.8 |
| DSB | 10.0 | 96.1 | 9.5 | 74.8 |
| VERICOT-NoPrem | 3.3 | 100 | 3.3 | 75.8 |
| VERICOT | 45.2 | 94.1 | **42.5** | 75.8 |
| **BioASQ** | | | | |
| ER | 1.5 | 80.0 | 1.2 | 81.4 |
| DSB | 5.9 | 72.2 | 4.2 | 75.7 |
| VERICOT-NoPrem | 2.9 | 57.1 | 1.6 | 81.4 |
| VERICOT | 25.3 | 84.3 | **21.3** | 81.4 |
| **Legalbench-SARA** | | | | |
| ER | 6.8 | 92.0 | 6.3 | 80.0 |
| DSB | 4.8 | 94.1 | 4.5 | 77.7 |
| VERICOT-NoPrem | 0.6 | 50.0 | 0.3 | 80.0 |
| VERICOT | 15.2 | 87.0 | **13.2** | 80.0 |

Table 1: Verification results across benchmarks. For each dataset, we report verification pass rate (%), verifier precision (%), verified correct answer rate (VCAR, %), and task accuracy (%, i.e., correct answer rate). VERICOT consistently achieves the highest values across the main outcome metrics (pass rate and VCAR).

| Category | Score |
|---|---|
| **Legalbench-SARA** | |
| Grounded contextual premise | 93.5 |
| Acceptable commonsense premise | 83.9 |
| Necessary commonsense premise | 77.0 |
| **BioASQ** | |
| Grounded contextual premise | 87.3 |
| Acceptable commonsense premise | 93.0 |
| Necessary commonsense premise | 81.1 |
| **ProofWriter** | |
| Grounded contextual premise | 96.4 |
| Acceptable commonsense premise | 90.5 |
| Necessary commonsense premise | 77.4 |

Table 2: Extended analysis of LLM-as-Judge premise evaluation breakdown. The generated premises of all types are highly acceptable under LLMaJ.

## 3.1 DATASETS

We consider three datasets. **ProofWriter** (Tafjord et al., 2021) contains small rulebases of facts and rules. Each rulebase has a set of questions which can either be proven true or false using proofs of various depths. **LegalBench** (Guha et al., 2023) is a benchmark designed to evaluate the legal reasoning capabilities of LLMs crowdsourced from legal professional; we use the **SARA** (Holzenberger et al., 2020) subset from LegalBench, which evaluates statutory reasoning in tax law. **BioASQ** (Nentidis et al., 2023; 2024) is a series of annual open challenges focused on advancing biomedical semantic indexing and QA, specifically we use task b, which is biomedical QA with PubMed abstracts as context. Appendix Table 6 shows data statistics.

## 3.2 BASELINES

We compare VERICOT against three baselines using Claude-3.5-Sonnet-V2. **Explanation-Refiner (ER)** (Quan et al., 2024b): an iterative auto-formalization and refinement approach guided by a theorem prover, originally proposed for NLI tasks. In our adaptation, we treat the CoT steps as the explanation, the question as the premise, and the model-generated answer as the hypothesis. **Direct SMT Baseline (DSB)**: A direct formalization approach that decomposes reasoning steps with designated types and generates SMT-LIB expressions simultaneously via few-shot prompting. For reasoning steps based on document, background, or commonsense premises, only a consistency check is applied; for inference steps, an entailment check is performed. **VERICOT without Explicit Premise Generation (VERICOT-NoPrem)**: A variant of our VERICOT pipeline with generating intermediate premises turned off. Verification is performed using the same check protocol as in the Direct SMT Baseline.

## 3.3 VERICOT AS AN EFFECTIVE VERIFICATION ALGORITHM

We first evaluate VERICOT as a verification framework for CoTs, demonstrating that it achieves high verification performance by capturing a larger proportion of verifiable reasoning with strong precision and introduces an innovative mechanism for analyzing the premise grounding of reasoning steps.

**Direct evaluation of VERICOT.** We evaluate verification methods using four complementary metrics: verification pass rate, verifier precision, verified correct answer rate (VCAR), and overall task accuracy. Following prior CoT-verification work such as Explanation-Refiner (Quan et al., 2024b), pass rate measures the fraction of model-generated CoTs that a verifier judges as valid, capturing verification coverage rather than task-solving ability. Precision captures the proportion of correct answers among verified CoTs, directly reflecting the reliability of VERICOT decisions.

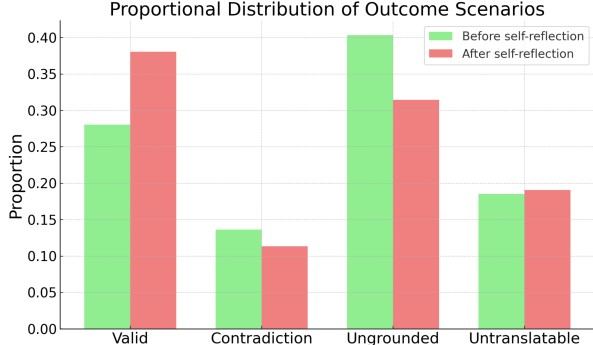

Figure 2: Proportional distribution of outcome scenarios before and after self-reflection (§3.4) under VERICOT. Categories include successful verification (Valid) and failure cases: Contradiction, Ungrounded, Untranslatable as described in §2. Self-reflection significantly reduce errors.

| Dataset | Accurate Translation | Grounded Contextual | Acceptable Commonsense | Correct LLMaJ |
|---------|---------------------|---------------------|------------------------|---------------|
| BioASQ | 0.80 | 0.87 | 0.91 | 0.97 |
| SARA | 0.86 | 0.92 | 0.89 | 0.95 |
| ProofWriter | 0.87 | 0.95 | 0.95 | 1.00 |

Table 3: Evaluation of translation and premise quality across datasets. We report the proportion of accurate translations, grounded contextual premises, acceptable commonsense premises, and correct LLMaJ outcomes.

VCAR combines these two by quantifying the overall fraction of CoTs that are both verified and correct. The metric penalizes both the verifier that is too lenient (high pass rate, low precision) and the verifier that is overly selective and only accepts trivial cases (low pass rate, high precision). Finally, task accuracy corresponds to the final task-level correctness rate. As shown in Table 1, VERICOT achieves the highest pass rates across all benchmarks, leading to the strongest verified correct answer rate. Importantly, its precision is consistently high, exceeding the final task accuracy, demonstrating that VERICOT-validated CoTs provide a more reliable signal of correctness than the raw CoTs. These results highlight VERICOT as an effective and accurate verification algorithm that achieves broader coverage with better reliability.

**Detailed analysis of instances that fail the VERICOT verification.** We first conduct a quantitative analysis by presenting the proportional distribution of each possible outcome scenario under VERICOT before and after self-reflection (introduced below in §3.4). Ungrounded error is the most prominent, highlighting that CoT often overgenerates assumptions, producing reasoning steps that appear plausible but unsupported. Self-reflection significantly increases the proportion of valid outcomes while reducing ungrounded and contradiction errors. The rate of translation-related errors remains largely unchanged. Additionally, we provide three illustrative examples in §A.4, each corresponding to one type of failure case. For each example, we present the updated CoT along with its successful verification after self-reflection, as detailed in §3.4.

**Quantitative analysis of NL premise quality.** We present the results of VERICOT premise evaluation as described in §2.4. We use LLMaJ to assess the quality of the generated premises: (1) grounded contextual premise: whether a contextual premise is correctly grounded in the source document or question, and (2) acceptable commonsense premise: whether a commonsense premise is reasonable given the context and the targeted reasoning step. In addition, we evaluate the necessity of each commonsense premise (necessary commonsense premise), i.e., whether it is required to support the corresponding reasoning step. The results in Table 2 confirm the high quality of premises identified by VERICOT. We present examples of different NL premises in Appendix A.3. A key strength of VERICOT is reliably locating reasoning steps in context or commonsense, making implicit premises explicit. For users to evaluate grounding, the challenge shifts from locating supporting premises to evaluating the quality of the premises that VERICOT has already identified as shown in Appendix Tables 5, 6 and 7.

**Human Evaluation of VERICOT.** We conduct a human analysis by sampling 20 examples per dataset, evenly across the four labels: verifiable, ungrounded, untranslatable, and contradiction. We assess whether each logic translation is accurate, faithfully captures all necessary details, and

| Method | Pass Rate | ΔPR | Precision | ΔPrecision | VCAR | ΔVCAR | Task Accuracy |
|---|---|---|---|---|---|---|---|
| **ProofWriter** | | | | | | | |
| ER | 21.5 | +6.7 | 80.2 | -3.1 | 17.2 | +4.9 | 75.8 |
| DSB | 12.3 | +2.3 | 90.6 | -5.5 | 11.1 | +1.6 | 73.2 |
| VERICOT-NoPrem | 30.7 | +27.4 | 88.8 | -11.2 | 27.3 | +24.0 | 72.4 |
| VERICOT-Base | 60.1 | +14.9 | 90.1 | -4.0 | 54.1 | +11.6 | **74.1** |
| VERICOT-w LLMaJ | **60.6** | +15.4 | 90.4 | -3.7 | **54.8** | +12.3 | 73.7 |
| **BioASQ** | | | | | | | |
| ER | 3.8 | +2.3 | 84.6 | +4.6 | 3.2 | +2.0 | 81.4 |
| DSB | 8.5 | +2.6 | 76.9 | +4.7 | 6.5 | +2.3 | 78.8 |
| VERICOT-NoPrem | 29.7 | +26.8 | 81.6 | +24.5 | 24.2 | +22.6 | 79.8 |
| VERICOT-Base | 33.4 | +8.1 | 86.3 | +2.0 | 28.8 | +7.5 | **82.0** |
| VERICOT-w LLMaJ | **36.9** | +11.6 | 83.8 | -0.5 | **30.9** | +9.6 | 80.5 |
| **Legalbench-SARA** | | | | | | | |
| ER | 19.6 | +12.8 | 91.6 | -0.4 | 18.0 | +11.7 | 80.0 |
| DSB | 9.0 | +5.2 | 93.8 | -0.3 | 8.4 | +3.9 | 80.2 |
| VERICOT-NoPrem | 16.1 | +15.5 | 84.2 | +34.2 | 13.6 | +13.3 | 80.3 |
| VERICOT-Base | **25.2** | +10.0 | 85.5 | -1.5 | **21.6** | +8.4 | **81.4** |
| VERICOT-w LLMaJ | 23.0 | +7.8 | 85.5 | -1.5 | 19.7 | +6.5 | 81.0 |

Table 4: Verification results after self-reflection. In addition to the metrics in Table 1, we report absolute changes compared to pre-reflection (Δ). VERICOT-Base uses only verifier signals without LLMaJ evaluation, while VERICOT-w LLMaJ leverages the full set of signals for self-reflection. VERICOT(Base,w LLMaJ) achieves the strongest improvements (VCAR and accuracy) across all benchmarks while maintaining precision at a similar or higher level compared to baselines.

| Method | Pass Rate | Precision | VCAR | Task Accuracy |
|---|---|---|---|---|
| **BioASQ** | | | | |
| Qwen2.5-7B-Instruct (direct) | 22.8 | 83.1 | 18.9 | 77.4 |
| + SFT w Random Distilled CoTs | 22.9 | 85.7 | 19.6 | 78.5 |
| + SFT w Verified CoTs | 22.9 | 85.5 | 19.6 | 79.7 |
| + SFT w Verified CoTs + DPO | **26.8** | 83.5 | **22.3** | 79.4 |
| **ProofWriter** | | | | |
| Qwen2.5-7B-Instruct (direct) | 21.8 | 76.7 | 16.7 | 47.5 |
| + SFT w Random Distilled CoTs | 24.6 | 71.4 | 17.6 | 47.3 |
| + SFT w Verified CoTs | 23.2 | 81.5 | 18.9 | 51.1 |
| + SFT w Verified CoTs + DPO | **27.8** | 82.8 | **23.0** | 51.8 |

Table 5: SFT and DPO results on BioASQ and ProofWriter. We report pass rate (%), verifier precision (%), verified correct answer rate (VCAR, %), and task accuracy (%). Strategies include the base model (direct), fine-tuning with random distilled CoTs, verified CoTs, and DPO with verified CoTs.

preserves the correct logical relationships. We also evaluate the quality of the generated premises: (1) whether a contextual premise is correctly grounded in the source document or question, and (2) whether a commonsense premise is acceptable and reasonable given the context and the targeted reasoning step. Finally, we check whether LLMaJ's judgments align with human annotations. As shown in Table 3, across all criteria, we observe consistently high performance. The autoformalizer exhibits a relatively low error rate, under 20%, which is already quite strong for NL-to-FoL translation. LLMaJ decisions on premises are aligned with human judgements 95-100% of the time. We also conducted a human analysis on the model's CoT decomposition. Across 60 examples (20 per dataset), we found that 96.7% of decompositions accurately preserved the information in the original CoT.

## 3.4 APPLICATIONS OF VERICOT'S VERIFICATION SIGNALS

The verification signals produced by VERICOT provide structured feedback that improves CoT reasoning along three dimensions: user transparency, inference-time self-reflection, and fine-tuning. First, by auto-formalizing CoTs and making premises explicit, VERICOT increases transparency, enabling users to directly and more effectively inspect the reasoning chain. Second, VERICOT supports self-reflection at inference time by exposing granular errors in faulty CoTs, thereby providing

actionable signals that help correct CoT. Third, the same structured signals can be incorporated into supervised fine-tuning and direct preference optimization (DPO) (Rafailov et al., 2024), guiding models toward producing more verifiable CoTs and ultimately achieving higher task accuracy.

**Inference-time Self-reflection.** We first evaluate whether VERICOT feedback can be used to guide models in producing better CoTs at inference time. Our process is as follows: if a CoT fails to pass the verification check, we prompt the model to self-correct its reasoning. We adopt two variants (Base, w LLMaJ): Base uses only verifier signals without LLMaJ premise evaluation, while VERICOT-w LLMaJ leverages the full set of signals for self-reflection. Given all relevant information for every step, including the original reasoning steps, any added premises, the corresponding formalizations, errors, check results (with execution results and the model's parameter assignment), and optional LLMaJ premise evaluation, the model is prompted to revise its CoT reasoning during inference. This generates an updated CoT, which is then re-evaluated.

Table 4 reports verification pass rates, verifier precision, verified correct answer rates (VCAR), and final task accuracy after self-reflection, together with the corresponding improvements relative to pre-reflection values (ΔPR, ΔPrecision, and ΔVCAR). Across all benchmarks, VERICOT with self-reflection yields consistent gains in both coverage ( average +12.3%(absolute)/ +46.4% (relative)) and verified correct answer rate ( average +9.5%(absolute)/ +41.1% (relative)), indicating that refinement encourages the generation of reasoning that is more frequently verifiable and leads to correct answers. VERICOT achieves the highest absolute pass rates and verified correct answer rate, while maintaining a similar level of precision and slightly better final task accuracy. These results highlight the effectiveness of VERICOT's verification signals in guiding self-reflection. We also report a variant (w LLMaJ) that adds additional LLMaJ evaluation signal for self-reflection; however, it yields only a marginal improvement over the Base baseline. This is expected as the verifier results provide much more informative error signals.

**Supervised Fine-tuning.** To harness the benefits of verification for model improvement, we build high-fidelity datasets consisting of verified, logically consistent CoT examples (pass both verifier checks and LLMaJ evaluation) and apply them for supervised fine-tuning (SFT). We start from the Qwen2.5-7B-Instruct model and distill supervision signals from Claude-3.5-Sonnet-v2. Table 5 benchmarks different distillation strategies to evaluate the role of verification in data curation. Specifically, we compare (i) the baseline performance of Qwen2.5-7B-Instruct, (ii) SFT with randomly sampled distilled CoTs, (iii) SFT with distilled CoTs that pass VERICOT verification. As shown in Table 5, requiring CoTs to pass verification (ii vs. iii) yields an average 3% improvement in final accuracy. This is because the verifier's higher precision ensures that verified CoTs contain a greater proportion of correct answers than randomly selected ones. This highlights VERICOT's signal as particularly valuable, as it can distill better data when gold answers are unavailable.

**Preference Fine-tuning.** The verification signals from each reasoning step can further be utilized as pairwise reward signals during preference fine-tuning (PFT) with DPO to enable more reliable CoT generation. We randomly resample CoTs for examples that initially pass VERICOT verification, keep those that fail in the new attempt, and then construct chosen/rejected CoT pairs, where the chosen/rejected one passes/fails verification. When applying DPO on top of SFT , we observed two key improvements across all datasets as shown in Table 5: the verification pass rate increased by 4.3% (an 18.4% relative improvement), while the verified correct answer rate improved by 3.4% (a 17.7% relative gain). This demonstrates that preference fine-tuning on data generated by VERICOT is especially helpful for generating verified CoTs because the pairwise reward signal provides a stronger supervision signal to distinguish correct reasoning traces from incorrect ones.

## 4 RELATED WORK

**Solver-based Verification of Natural Language Reasoning.** The most relevant work to ours is Quan et al. (2024b), where a neuro-symbolic pipeline is introduced to formalize natural-language explanations (for natural language inference) into logical forms using LLMs, and then performs theorem-prover–based verification and refinement of those explanations. While Quan et al. (2024b) focuses on verifying and refining explanations post hoc for NLI tasks, our framework not only translates and verifies each step of a CoT into first-order logic (SMT-LIB) but also grounds the reasoning chains in context and is tailored for multi-step reasoning beyond NLI explanations. Additionally, works like Ling et al. (2023) and Vacareanu et al. (2024) use LLMs to assess the logical consistency

of each reasoning step, whereas our method adds formal solver checks, context grounding, and correction, yielding more objective verification than LLM-only pipelines.

**Solver-empowered Logical Reasoning in LLMs.** Recent works have explored solver-assisted QA with LLMs (Pan et al., 2023; Poesia et al., 2024; Ye et al., 2023; Kirtania et al., 2024; Ganguly et al., 2024; Arakelyan et al., 2025). Specifically, Pan et al. (2023) translates NL into logic and uses solvers with iterative repair; building on this, Kirtania et al. (2024) further uses the ability of LLMs to do pairwise comparisons, allowing the evaluation of the refinements suggested by the LLM; Ye et al. (2023) prompts LLMs to produce declarative constraints for SAT solving; Poesia et al. (2024) constrains LLM outputs via symbolic state tracking. Xu et al. (2024); Arakelyan et al. (2025) propose symbolic CoT-style methods that generate formal reasoning steps to improve faithfulness to context and logical constraints without relying on external solvers.Ganguly et al. (2024) uses the LLM to generate a formal "thought program" in its intermediate JSON-based Domain-Specific Language (DSL), which serves as the verifiable logical representation of the reasoning step. All above works have LLMs generate solver code during reasoning with a focus on task success, i.e., generating reasoning traces that are mostly NL but with verified components that help maximize benchmark performance. VERICOT, however, takes a CoT as input and determines whether this reasoning is logically valid, imposing a strict constraint on the CoT's verifiability. We differ by combining context-grounded `SMT-LIB` translation with step-wise verification and using solver feedback for on-the-fly reasoning correction. Our RL setup is closest to Leang et al. (2025). They ground judgments in Lean's mathlib (math-focused) and demonstrate the effectiveness of "Theorem Prover-as-a-Judge", whereas VERICOT grounds statements in premises inferred from natural-language context, making it applicable across general NL domains.

**Structured Explanation Generation for LLM Reasoning** Our work relates to growing literature on surfacing the underlying premises and stepwise reasoning for task-specific conclusions. VERI-COT surfaces NL premises that, when formalized, provide a directed, tree-like logical basis for a symbolic conclusion; this is similar to the fully NL entailment tree structure (Dalvi et al., 2021; Bostrom et al., 2021) which uses NL textual entailment as the "logical glue" between each layer of inferences. Versions of entailment tree-generating algorithms such as Entailer (Tafjord et al., 2022), ADGV (Sprague et al., 2022), and NELLIE (Weir et al., 2024) map from task answers to entailment trees rooted in underlying conjunctive premises. Maieutic Prompting (Jung et al., 2022) produces related structures rooted in underlying arguments for/against believing recursive inferences.

## 5 LIMITATIONS

As both autoformalization and premise inference rely on LLMs, it is possible that either step can be incorrect (either by mis-translating the CoT into an unrepresentative formula, or by introducing a premise that is not well-founded). A formalization may be incorrect simply because the LLM used made a mistake during translation, or because the text to be translated fundamentally cannot be represented in the subset of `SMT-LIB` that we support. As a result, while VERICOT can prove that the formalization of the Chain-of-Thought necessarily follows from the inferred premises, it cannot guarantee that the NL CoT or the premises are correct.

Potential future work includes: (1) reducing the number of LLM calls to lower end-to-end latency of VERICOT; (2) improving robustness to autoformalization errors through tighter constraints and more explicit generation instructions, together with more effective translation–verification feedback; and (3) extending VERICOT to more complex domains that require richer reasoning structures, such as temporal or procedural reasoning.

## 6 CONCLUSION

In this paper, we present a neuro-symbolic framework that validates Chain-of-Thought reasoning by autoformalizing each intermediate step into first-order logic and grounding it in formal premises inferred from NL context or commonsense knowledge. Across the ProofWriter, LegalBench-SARA, and BioASQ datasets, VERICOT reliably detects ungrounded or incorrect reasoning in CoT traces, which we show is a strong predictor of final answer correctness. We further show that these verification signals can guide models to self-correct during inference and through supervised fine-tuning and preference fine-tuning for better CoT reasoning.

ACKNOWLEDGMENTS

This work was conducted during Yu Feng's internship at Amazon Web Services (AWS). We sincerely thank Nick Feng and Dan Roth for their valuable feedback, and we are also grateful to the ICLR reviewers and editors for their insightful and constructive comments.

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

## A   APPENDIX

### A.1   USE OF LARGE LANGUAGE MODELS (LLMS)

Large Language Models (LLMs) are used in the following scenarios:

Grammar Checking: we use Grammarly only for grammar checks, but this is an AI-powered writing assistant.

Formatting Assistance: we use Claude 3.7 Sonnet to adjust figure and table formatting for consistency and readability.

No LLMs contributed to research ideation or drafting of substantive scientific content. The authors take full responsibility for all content presented in the paper.

### A.2   DATA STATISTICS

| Dataset | Train Size | Test Size | Description |
|---|---|---|---|
| **ProofWriter** | 5000 | 400 | Randomly sampled from the OWA-Depth-5 and CWA-Depth-5 subsets. Contains only entailment/contradiction labels (no "Unknown" answers). Source: https://huggingface.co/datasets/renma/ProofWriter. |
| **BioASQ** | 5,049 | 340 | Restricted to Task 12b: Test Results of Phase B. |
| **LegalBench (SARA)** | - | 367 | 272 instances from SARA Entailment and 95 instances from SARA Numeric. |

Table 6: Dataset statistics and descriptions.

### A.3   EXAMPLES OF DIFFERENT PREMISES UNDER VERICOT

We present two qualitative examples to demonstrate different type of generated premises. In the example in Appendix Fig. 3, all contextual premises are grounded and all commonsense premises are acceptable and necessary. In the example in Appendix Fig. 4, all contextual premises are grounded and all commonsense premises are acceptable but unnecessary.

| Question context | Output premise list |
|---|---|
| **Question:** Alice's taxable income for the year 2017 is $25,561. How much taxes does Alice have to pay for the year 2017 under Statute section 1(a)(i)? 
 **Context (partially omitted):** ... Statute section 1(a)(i): 15% of taxable income if the taxable income is not over $36,900. ... 
 **Step-wise CoT:** 
 1.  Statute specifies a tax rate of 15% for taxable income not exceeding $36,900. 
 2.  Alice's taxable income is $25,561. 
 3.  Since $25,561 < $36,900, Alice falls in the 15% bracket. 
 4.  Tax = $25,561 × 0.15 = $3,834.15, rounding to $3,834. 
 5.  Alice must pay $3,834. | **Document:** 
 1.  The tax rate is exactly 15% for taxable income not over $36,900. 

 **Question:** 
 1.  Alice's taxable income for the year 2017 is $25,561. 

 **Commonsense:** 
 1.  The tax amount is calculated by multiplying the taxable income by the tax rate percentage divided by 100. 
 2.  The rounded tax amount is the exact tax amount rounded to the nearest whole dollar, where amounts with cents less than 50 round down and amounts with 50 cents or more round up. |

Figure 3: Example 1 from SARA: grounded reasoning and extracted premises.

### A.4   EXAMPLES OF FAILURE CASES UNDER VERICOT

We present three illustrative cases of different types of errors, each of which, after self-reflection, passes verification. Specifically, these include (1) an ungrounded example from ProofWriter as shown

| Question context | Output premise list |
|---|---|
| **Question:**
Based on the above information, is the following statement true, false, or unknown?
*Charlie is blue.*
**Context:**
Bob is big. Bob is nice. Bob is smart. Charlie is nice. Dave is nice. Erin is big. Erin is blue. Erin is furry. Erin is quiet. Erin is round. Erin is smart. All nice things are big. All furry things are blue. Blue, furry things are quiet. If something is round and nice then it is blue. If Bob is round then Bob is blue. Big, quiet things are furry. If something is blue and furry then it is smart. If Charlie is nice and Charlie is big then Charlie is quiet.
**Step-wise CoT:**
1. Charlie is nice.
2. Since all nice things are big, and Charlie is nice, Charlie must be big.
3. Since Charlie is both nice and big, and there is a rule stating "If Charlie is nice and Charlie is big then Charlie is quiet", Charlie must be quiet.
4. Since Charlie is both big and quiet, and there is a rule stating "Big, quiet things are furry", Charlie must be furry.
5. Since Charlie is furry, and there is a rule stating "All furry things are blue", Charlie must be blue.
6. Therefore, the statement "Charlie is blue" is true. | **Document:**

1. Charlie is nice.
2. All nice things are big.
3. If Charlie is nice and Charlie is big then Charlie is quiet.
4. Big, quiet things are furry.
5. All furry things are blue.

**Commonsense:**

1. Being big is a property that some things have and others don't.
2. Being quiet is a property that some things have and others don't. |

Figure 4: Example from ProofWriter: grounded reasoning and extracted premises for "Charlie is blue".

in Appendix Table 5 (refer to full details in A.4.1): the model initially is over-claiming. During self-reflection, however, it is able to identify the overclaimed portion, rephrase it appropriately, and thus generate a verified CoT.

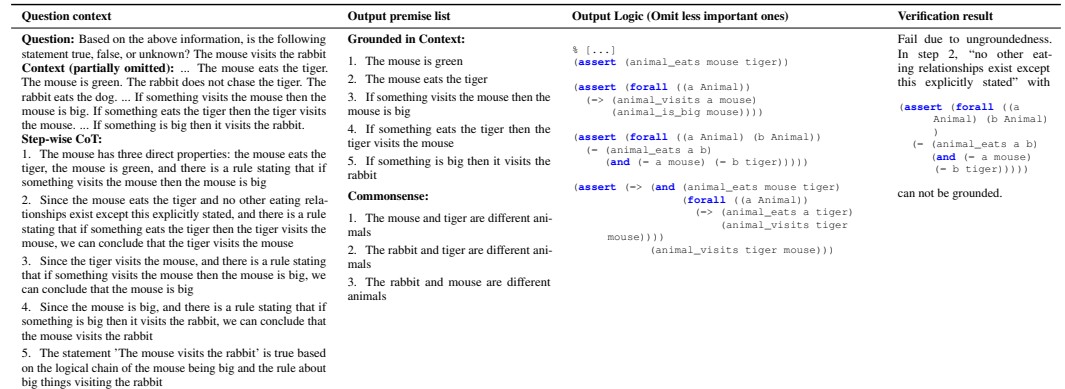

Figure 5: Highlights of an ungrounded example from Proofwriter before self-reflection.

(2) a contradiction example from BioASQ as shown in Appendix Table 6 (refer to full details in A.4.2): the reasoning contains a subtle logical flaw that is difficult to detect. Our VeriCoT framework successfully identifies this issue and prompts the model to explore an alternative reasoning path during self-reflection, ultimately resulting in a verified CoT.

(3) an untranslatable example from LegalBench-SARA as shown in Appendix Table 7 (refer to full details in A.4.3): when the translation error is presented to the model for self-reflection, the model recognizes that the erroneous reasoning step is unnecessary for the final solution. By removing this step, the model produces a verified CoT.

| Question context | Output premise list | Output Logic (Omit less important ones) | Verification result |
|---|---|---|---|
| **Question:** Can Connexin hemi channels be used for drug delivery?
**Context omitted**
**Step-wise CoT:**
1. Connexin hemichannels are channels in cell membranes that can allow molecules to pass through and are part of gap junction channels.
2. Connexin hemichannels allow for the non-selective passage of molecules up to 1,000 Da.
3. The gap junction channel can transport macro polymer therapeutic molecules, which is evidence for drug delivery potential.
4. Therefore, Connexin hemi channels are also permeable to therapeutic macromolecules, makes them highly attractive targets for delivering drugs. | **Grounded in Context:**
1. Connexin hemichannels are channels in membranes that pass molecules.
2. Connexin hemichannels are part of gap junction.
3. Connexin hemichannels allow for the non-selective passage of molecules up to 1,000 Da.
4. The ability to deliver large polymeric therapeutic macromolecules via the gap junction route provides evidence for drug delivery potential.

**Commonsense:**
1. Any macro polymeric therapeutic molecule has a molecular weight $> 1,000$ Da. | `% [...]`
`(assert (forall ((m Molecule) (c Channel))`
`  (=> (and (IsConnexinHemichannel c)`
`          (CanEnterViaChannel m c))`
`      (<= (MolecularWeight m) 1000))))`
`(assert (forall ((m Molecule))`
`  (=> (IsMacroPolymerTherapeutic m)`
`      (> (MolecularWeight m) 1000))))`
`(assert (forall ((c Channel) (m Molecule))`
`  (=> (and (IsGapJunctionChannel c)`
`          (IsMacroPolymerTherapeutic m))`
`      (and (CanEnterViaChannel m c)`
`          (=> (CanEnterViaChannel m c)`
`              (HasDrugDeliveryPotential c)))`
`  )))`
`(assert (exists ((c Channel) (m Molecule))`
`  (and (IsConnexinHemichannel c)`
`      (IsMacroPolymerTherapeutic m)`
`      (CanEnterViaChannel m c))))` | Failed due to contradiction. In step 4, the axioms show macro-polymer molecules are $> 1000$ Da, but connexin hemichannels only permit molecules $\leq 1000$ Da. Thus, the same molecule would need to be both $\leq 1000$ and $> 1000$, which is impossible. |

Figure 6: Highlights of a contradiction example from BioASQ before self-reflection.

| Question context | Output premise list | Output Logic (Omit less important ones) | Verification result |
|---|---|---|---|
| **Question:** Determine whether the following statements are entailed under the statute. Reply with either: Entailment, Contradiction.
**Context:** Alice has paid $3200 to Bob for agricultural labor done from Feb 1st, 2017 to Sep 2nd, 2017. Alice is an American employer. Section 3306(c)(1) applies to Alice employing Bob for the year 2017.
**CoT:**
1. Section 3306(c)(1) of the statute addresses agricultural labor
2. The work performed by Bob for Alice is specifically agricultural labor
3. The basic scope of Section 3306(c)(1) covers all agricultural labor
4. While Alice is an American employer and there might be jurisdictional implications, the statute fragment does not provide information about geographical limitations
5. Bob's employment falls under the basic scope of Section 3306(c)(1) | **Grounded in Context:**
1. Section 3306(c)(1) applies to agricultural labor
2. Alice employed Bob to perform agricultural labor from February 1st to September 2nd, 2017, as evidenced by Alice's payment of $3200 for this work | `% [...]`
`(assert`
`  (and`
`    (employs alice bob)`
`    (performs_agricultural_labor bob)`
`    (paid alice bob agricultural_labor`
`      3200 20170201 20170902)))`
`(assert`
`  (forall ((p Person))`
`    (=> (performs_agricultural_labor p)`
`        (covered_by_section_3306c1 p)))`
`  )`
`(assert   (covered_by_section_3306c1 bob`
`    ))` | Fail due to untranslatability. In step 4, the phrase "there might be jurisdictional implication" cannot be rendered under SMI-LIB rules because it expresses a possibility rather than a definitive statement. |

Figure 7: Highlights of an untranslatable example from LegalBench-SARA before self-reflection.

### A.4.1 Ungrounded example from ProofWriter

---

**Ungrounded example from ProofWriter**

**Reference information:** The dog does not eat the rabbit. The mouse eats the tiger. The mouse is green. The rabbit does not chase the tiger. The rabbit eats the dog. The rabbit is young. The tiger is green. If something visits the mouse then the mouse is big. If something eats the tiger then the tiger visits the mouse. If the dog is young and the dog does not visit the mouse then the mouse does not visit the rabbit. If something is green and it chases the rabbit then the rabbit does not eat the mouse. If something is green then it visits the dog. If something visits the rabbit and the rabbit is young then it is round. If something is round and it visits the dog then it chases the tiger. If something is big then it visits the rabbit.

**Question:** Based on the above information, is the following statement true, false, or unknown? The mouse visits the rabbit.

**Answer:** True

**Original CoT:**
Let me solve this step by step.

1. First, let's identify what we know about the mouse:
* The mouse eats the tiger
* The mouse is green
* If something visits the mouse then the mouse is big
2. Since the mouse eats the tiger and and no other eating relationships exist except this explicitly stated, according to the rule:
* "If something eats the tiger then the tiger visits the mouse"
* Therefore, the tiger visits the mouse
3. Since the tiger visits the mouse, according to the rule:
* "If something visits the mouse then the mouse is big"
* Therefore, the mouse is big
4. Now, since the mouse is big, according to the rule:
* "If something is big then it visits the rabbit"
* Therefore, the mouse visits the rabbit
5. We can directly conclude that the mouse visits the rabbit based on these logical steps.
Therefore, the statement "The mouse visits the rabbit" is TRUE.

---

VERICOT Output before self-reflection:

---

**Decomposed CoT Steps**

1. The mouse has three direct properties: the mouse eats the tiger, the mouse is green, and there is a rule stating that if something visits the mouse then the mouse is big
2. Since the mouse eats the tiger and no other eating relationships exist except this explicitly stated, and there is a rule stating that if something eats the tiger then the tiger visits the mouse, we can conclude that the tiger visits the mouse
3. Since the tiger visits the mouse, and there is a rule stating that if something visits the mouse then the mouse is big, we can conclude that the mouse is big
4. Since the mouse is big, and there is a rule stating that if something is big then it visits the rabbit, we can conclude that the mouse visits the rabbit
5. The statement 'The mouse visits the rabbit' is true based on the logical chain of the mouse being big and the rule about big things visiting the rabbit

---

**Step 1**

**Step Context:** The mouse has three direct properties: the mouse eats the tiger, the mouse is green, and there is a rule stating that if something visits the mouse then the mouse is big.

---

```
; ========================
; Schema
; ========================
; represents an animal entity
(declare-sort Animal)
; constant representing the mouse
(declare-const mouse Animal)
; constant representing the tiger
(declare-const tiger Animal)
; true if one animal eats another
(declare-fun animal_eats (Animal Animal) Bool)
; true if an animal is green
(declare-fun animal_is_green (Animal) Bool)
; true if an animal is big
(declare-fun animal_is_big (Animal) Bool)
; true if one animal visits another
(declare-fun animal_visits (Animal Animal) Bool)

; ========================
; Added premises
; ========================
; The mouse is green (document)
(assert (animal_is_green mouse))
; The mouse and tiger are different animals (commonsense)
(assert (not (= mouse tiger)))
; The mouse eats the tiger (document)
(assert (animal_eats mouse tiger))
; If something visits the mouse then the mouse is big (doccument)
(assert (forall ((a Animal))
  (=> (animal_visits a mouse)
      (animal_is_big mouse))))

; ========================
; Step logic
; ========================
; mouse eats tiger
; mouse is green
; if animal visits mouse then mouse is big
(assert (animal_eats mouse tiger))
(assert (animal_is_green mouse))
(assert (forall ((a Animal))
  (=> (animal_visits a mouse)
      (animal_is_big mouse))))
```

**Evaluation:** Pass

---

Step 2

**Step Context:** Since the mouse eats the tiger and no other eating relationships exist except this explicitly stated, and also there is a rule stating that if something eats the tiger then the tiger visits the mouse, we can conclude that the tiger visits the mouse.

```
; ======================
; Schema
; ======================
; (none new)

; ======================
; Added premises
; ======================
; If something eats the tiger then the tiger visits the mouse (document)
(assert (forall ((a Animal))
  (=> (animal_eats a tiger)
      (animal_visits tiger mouse))))

; ======================
; Step logic
; ======================
; The mouse eats the tiger and no other eating relationships exist except those
;     explicitly stated
; if if mouse eats tiger and if a eats tiger then  tiger visits mouse
(assert (animal_eats mouse tiger))
(assert (forall ((a Animal) (b Animal))
  (= (animal_eats a b)
     (and (= a mouse) (= b tiger)))))
(assert (=> (and (animal_eats mouse tiger)
                 (forall ((a Animal))
                   (=> (animal_eats a tiger)
                       (animal_visits tiger mouse))))
            (animal_visits tiger mouse)))
```

**Evaluation:** Fail due to Ungrounded.
*Reason: "no other eating relationships exist except this explicitly stated" cannot be grounded.*

---

**Step 3**

**Step Context:** Since the tiger visits the mouse, and there is a rule stating that if something visits the mouse then the mouse is big, we can conclude that the mouse is big.

```
; ======================
; Schema
; ======================
; (none new)

; ======================
; Added premises
; ======================
; (none new)

; ======================
; Step logic
; ======================
; if tiger visits mouse and if any animal visits mouse then mouse is big
(assert (=>
  (and (animal_visits tiger mouse)
       (forall ((a Animal))
         (=> (animal_visits a mouse)
             (animal_is_big mouse))))
  (animal_is_big mouse)))
```

**Evaluation:** Pass

---

**Step 4**

**Step Context:** Since the mouse is big, and there is a rule stating that if something is big then it visits the rabbit, we can conclude that the mouse visits the rabbit.

```
; =======================
; Schema
; =======================
; constant representing the rabbit
(declare-const rabbit Animal)

; =======================
; Added premises
; =======================
; If something is big then it visits the rabbit (document)
(assert (forall ((a Animal))
  (=> (animal_is_big a)
      (animal_visits a rabbit))))

; =======================
; Step logic
; =======================
; if mouse big and if big then visits rabbit then mouse visits rabbit
(assert (=>
  (and (animal_is_big mouse)
       (forall ((a Animal))
         (=> (animal_is_big a)
             (animal_visits a rabbit))))
  (animal_visits mouse rabbit)))
```

**Evaluation:** Pass

**Step 5**

**Step Context:** The statement 'The mouse visits the rabbit' is true based on the logical chain of the mouse being big and the rule about big things visiting the rabbit.

```
; =======================
; Schema
; =======================
; (none new)

; =======================
; Added premises
; =======================
; (none new)

; =======================
; Step logic
; =======================
; if mouse is big and big animals visit rabbit then mouse visits rabbit
(assert (=>
  (and (animal_is_big mouse)
       (forall ((a Animal))
         (=> (animal_is_big a)
             (animal_visits a rabbit))))
  (animal_visits mouse rabbit)))
```

**Evaluation:** Pass

VERICOT Output after self-reflection:

**Decomposed CoT Steps**

1: From the reference information, we know three direct facts about the mouse: the mouse eats the tiger, the mouse is green, and there is a rule stating that if something visits the mouse then the mouse is big.
2: According to the reference information rule "If something eats the tiger then the tiger visits the mouse", and since we know the mouse eats the tiger, we can conclude that the tiger visits the mouse.
3: Since we have established that the tiger visits the mouse, and we know from the reference information that "If something visits the mouse then the mouse is big", we can conclude that the mouse is big.
4: The reference information states "If something is big then it visits the rabbit". Since we have established that the mouse is big, applying this rule means that the mouse visits the rabbit.

5: Therefore, based on the logical chain starting from the mouse eating the tiger, leading to the tiger visiting the mouse, making the mouse big, and ending with the rule about big things visiting the rabbit, we can conclude that the statement "The mouse visits the rabbit" is true.

**Step 1**

**Step Context:** From the reference information, we know three direct facts about the mouse: (1) The mouse eats the tiger, (2) The mouse is green, and (3) There is a rule stating that if something visits the mouse then the mouse is big.

```
; ========================
; Schema
; ========================
; represents an animal entity
(declare-sort Animal)
; constant representing the mouse
(declare-const mouse Animal)
; constant representing the tiger
(declare-const tiger Animal)
; true if one animal eats another
(declare-fun animal_eats (Animal Animal) Bool)
; true if an animal is green
(declare-fun animal_is_green (Animal) Bool)
; true if an animal is big
(declare-fun animal_is_big (Animal) Bool)
; true if one animal visits another
(declare-fun animal_visits (Animal Animal) Bool)

; ========================
; Added premises
; ========================
; The mouse is green (document)
(assert
  (animal_is_green mouse))

; The mouse and tiger are different animals (commonsense)
(assert
  (not
    (= mouse tiger)))
; The mouse eats the tiger (document)
; mouse eats tiger
(assert
  (animal_eats mouse tiger))

; If something visits the mouse then the mouse is big (document)
; if a visits mouse then mouse is big
(assert
  (forall
    ((a Animal))
      (=>
        (animal_visits a mouse)
        (animal_is_big mouse))))

; ========================
; Step logic
; ========================
; mouse eats tiger
; mouse is green
; if animal visits mouse then mouse is big
(assert
  (animal_eats mouse tiger))
(assert
  (animal_is_green mouse))
(assert
  (forall
    ((a Animal))
      (=>
        (animal_visits a mouse)
        (animal_is_big mouse))))
```

**Evaluation:** Pass

**Step 2**

**Step Context:** According to the reference information rule 'If something eats the tiger then the tiger visits the mouse', and since we know the mouse eats the tiger, we can conclude that the tiger visits the mouse.

```
; =======================
; Schema
; =======================
; (none new)

; =======================
; Added premises
; =======================
; If something eats the tiger then the tiger visits the mouse (document)
; if a eats tiger then tiger visits mouse
(assert
  (forall
    ((a Animal))
      (=>
        (animal_eats a tiger)
        (animal_visits tiger mouse))))

; =======================
; Step logic
; =======================
; if a eats tiger then tiger visits mouse
; if mouse eats tiger then tiger visits mouse
(assert
  (forall
    ((a Animal))
      (=>
        (animal_eats a tiger)
        (animal_visits tiger mouse))))
(assert
  (=>
    (animal_eats mouse tiger)
    (animal_visits tiger mouse)))
```

**Evaluation:** Pass

**Step 3**

**Step Context:** Since we have established that the tiger visits the mouse, and we know from the reference information that 'If something visits the mouse then the mouse is big', we can conclude that the mouse is big.

```
; =======================
; Schema
; =======================
; (none new)

; =======================
; Added premises
; =======================
; (none new)

; =======================
; Step logic
; =======================
; if tiger visits mouse then mouse is big
(assert
  (=>
    (animal_visits tiger mouse)
    (animal_is_big mouse)))
```

**Evaluation:** Pass

**Step 4**

**Step Context:** The reference information states 'If something is big then it visits the rabbit'. Since we have established that the mouse is big, applying this rule means that the mouse visits the rabbit.

```
; =======================
; Schema
; =======================
; constant representing the rabbit
(declare-const rabbit Animal)

; =======================
; Added premises
; =======================
;If something is big then it visits the rabbit (document)
; if a is big then a visits rabbit
(assert
  (forall
    ((a Animal))
    (=>
      (animal_is_big a)
      (animal_visits a rabbit))))

; =======================
; Step logic
; =======================
; if a is big then a visits rabbit
; if mouse is big then mouse visits rabbit
(assert
  (forall
    ((a Animal))
    (=>
      (animal_is_big a)
      (animal_visits a rabbit))))
(assert
  (=>
    (animal_is_big mouse)
    (animal_visits mouse rabbit)))
```

**Evaluation:** Pass

**Step 5**

**Step Context:** Therefore, based on the logical chain starting from the mouse eating the tiger, leading to the tiger visiting the mouse, making the mouse big, and ending with the rule about big things visiting the rabbit, we can conclude that the statement 'The mouse visits the rabbit' is true.

```
; ======================
; Schema
; ======================
; (none new)

; ======================
; Added premises
; ======================
; (none new)

; ======================
; Step logic
; ======================
; mouse eats tiger => tiger visits mouse
; tiger visits mouse => mouse is big
; mouse is big => mouse visits rabbit
; mouse visits rabbit
(assert
  (=>
    (animal_eats mouse tiger)
    (animal_visits tiger mouse)))
(assert
  (=>
    (animal_visits tiger mouse)
    (animal_is_big mouse)))
(assert
  (=>
    (animal_is_big mouse)
    (animal_visits mouse rabbit)))
(assert
  (animal_visits mouse rabbit))
```

**Evaluation:** Pass

### A.4.2  CONTRADICTION EXAMPLE FROM BIOASQ

**Contradiction example from BioASQ**

**Question:** Can Connexin hemi channels be used for drug delivery?

**Gold Answer:** Yes

**Context:**
The permeability of Cx43 channels to small molecules and macromolecules makes them highly attractive targets for delivering drugs directly into the cytoplasm. Cancer cells overexpressing Cx43 may be more permeable and sensitive to chemotherapeutics. In this context, certain channels lead to transitory plasma membrane permeability changes, such as pannexin, connexin hemichannels that are channels in membranes that pass molecules, TRPV1-4 and P2×7, which allow for the non-selective passage of molecules up to 1,000 Da.
Gap junction channels, composed of connexin proteins, provide a mechanism for direct transfer of small molecules across membranes, and recent evidence suggests that the transfer of larger, polymer-like molecules such as microRNAs may be possible.
Collectively, these results reveal that polymeric macromolecules can be delivered to cells via gap junctions, suggesting that the gap junction route can be used for the delivery of macro polymeric therapeutic molecules, which provides evidence for drug delivery potential.

**Original CoT:**
Let me break this down step by step.

1. First, let's understand what connexin hemichannels are:
* They are channels in cell membranes that can allow molecules to pass through.
* They are part of a gap junction.
2. Key characteristic of connexin hemichannels:
* They allow "non-selective passage of molecules up to 1,000 Da".
3. Evidence for drug delivery potential:
* Gap junction route can transport various types of macro polymeric therapeutic molecules.

**Conclusion:** Yes, connexin hemichannels can be used for drug delivery. The evidence strongly supports this, showing they can transport various types of therapeutic molecules, making them valuable targets for drug delivery applications.

**Decomposed CoT Steps**

1. Connexin hemichannels are channels in cell membranes that can allow molecules to pass through and are part of gap junction channels.

2. Connexin hemichannels allow for the non-selective passage of molecules up to 1,000 Da.

3. The gap junction route can transport macro polymer therapeutic molecules, which is evidence for drug delivery potential.

4. Therefore, Connexin hemi channels are permeable to therapeutic macromolecules, makes them highly attractive targets for delivering drugs.

---

**Step 1**

**Step Context:** Connexin hemichannels are channels in cell membranes that can allow molecules to pass through and are part of gap junction channels.

```
; =======================
; Schema
; =======================
; represents a molecule
(declare-sort Molecule)
; represents a channel
(declare-sort Channel)
; whether a channel is a connexin hemichannel
(declare-fun IsConnexinHemichannel (Channel) Bool)
; whether a channel is a gap junction channel
(declare-fun IsGapJunctionChannel (Channel) Bool)
; whether a channel is located in a cell membrane
(declare-fun IsInCellMembrane (Channel) Bool)
; whether an entity is considered a channel
(declare-fun IsChannel (Channel) Bool)
; whether a molecule can enter through a channel
(declare-fun CanEnterViaChannel (Molecule Channel) Bool)
; whether one channel is part of another channel
(declare-fun PartOf (Channel Channel) Bool)
; =======================
; Added premises
; =======================
; Connexin hemichannels are channels in membranes that pass molecules (document)
(assert (forall ((c Channel))
  (=> (IsConnexinHemichannel c)
      (and (IsChannel c)
           (IsInCellMembrane c)
           (exists ((m Molecule)) (CanEnterViaChannel m c))))))

; Connexin hemichannels are part of gap junction channels (document)
(assert (forall ((h Channel))
  (=> (IsConnexinHemichannel h)
      (exists ((g Channel))
        (and (IsGapJunctionChannel g)
             (PartOf h g))))))

; =======================
; Step logic
; =======================
; Connexin hemichannels are channels in membranes that pass molecules
; Connexin hemichannels are part of gap junction channels
(assert (forall ((c Channel))
  (=> (IsConnexinHemichannel c)
      (and (IsChannel c)
           (IsInCellMembrane c)
           (exists ((m Molecule)) (CanEnterViaChannel m c))))))
(assert (forall ((h Channel))
  (=> (IsConnexinHemichannel h)
      (exists ((g Channel))
        (and (IsGapJunctionChannel g)
             (PartOf h g))))))
```

**Evaluation:** Pass

---

**Step 2**

**Step Context:** Connexin hemichannels allow for the non-selective passage of molecules up to 1,000 Da.

```
; ======================
; Schema
; ======================
; returns the molecular weight of a molecule
(declare-fun MolecularWeight (Molecule) Real)

; ======================
; Added premises
; ======================
; Connexin hemichannels allow for the non-selective passage of molecules up to
    1,000 Da (document)
(assert (forall ((m Molecule) (c Channel))
  (=> (and (IsConnexinHemichannel c)
           (CanEnterViaChannel m c))
      (<= (MolecularWeight m) 1000))))

; ======================
; Step logic
; ======================
; If a molecule can enter via connexin channel, its weight must be <= 1000
(assert (forall ((m Molecule) (c Channel))
  (=> (and (IsConnexinHemichannel c)
           (CanEnterViaChannel m c))
      (<= (MolecularWeight m) 1000))))
```

**Evaluation:** Pass

Step 3

**Step Context:** Gap junction route can transport macro polymer therapeutic molecules, which is evidence for drug delivery potential.

```
; ======================
; Schema
; ======================
; whether a molecule is a macro polymer therapeutic molecule
(declare-fun IsMacroPolymerTherapeuticmolecule (Molecule) Bool)
; whether a channel has potential for drug delivery
(declare-fun HasDrugDeliveryPotential (Channel) Bool)
; ======================
; Added premises
; ======================
; The ability to deliver macro polymeric therapeutic molecules via the gap
      junction route,
; which provides evidence for drug delivery potential (document)
(assert
  (forall ((c Channel) (m Molecule))
    (=> (and (IsGapJunctionChannel c)
             (IsMacroPolymerTherapeuticmolecule m))
        (and (CanEnterViaChannel m c)
             (=> (CanEnterViaChannel m c)
                 (HasDrugDeliveryPotential c))))))

; Any marco polymeric therapeutic molecule has a molecular weight > 1,000 Da (
      commonsense)
(assert
  (forall ((m Molecule))
    (=> (IsMacroPolymerTherapeuticmolecule m)
        (> (MolecularWeight m) 1000))))

; ======================
; Step logic
; ======================
; If a gap junction channel transports large polymeric therapeutic
      macromolecules,
; then it has drug-delivery potential
(assert
  (forall ((c Channel) (m Molecule))
    (=> (and (IsGapJunctionChannel c)
             (IsMacroPolymerTherapeuticmolecule m))
        (and (CanEnterViaChannel m c)
             (=> (CanEnterViaChannel m c)
                 (HasDrugDeliveryPotential c))))))
```

**Evaluation:** Pass

Step 4

**Step Context:** Therefore, Connexin hemichannels are permeable to therapeutic macromolecules, making them valuable targets for delivering drugs.

```
; =======================
; Schema
; =======================
; (none new)

; =======================
; Added premises
; =======================
; (none new)

; =======================
; Step logic
; =======================
; Connexin hemichannels are permeable to therapeutic macromolecules
(assert
  (exists ((c Channel) (m Molecule))
    (and (IsConnexinHemichannel c)
         (IsMacroPolymerTherapeuticmolecule m)
         (CanEnterViaChannel m c))))

; Connexin hemichannels are permeable to therapeutic macromolecules,
; -> they are highly attractive drug-delivery targets
(assert
  (forall ((c Channel) (m Molecule))
    (=> (and (IsConnexinHemichannel c)
             (IsMacroPolymerTherapeuticmolecule m)
             (CanEnterViaChannel m c))
        (HasDrugDeliveryPotential c))))
```

**Evaluation:** Fail due to contradiction.
*Reason:* The axioms show macro-polymer molecules are $> 1000$ Da, but connexin hemichannels only permit molecules $\leq 1000$ Da. Thus, the same molecule would need to be both $\leq 1000$ and $> 1000$, which is impossible.

**Decomposed CoT Steps**

1. Connexin hemichannels are channels in cell membranes that can allow molecules to pass through and are part of gap junction channels.

2. Connexin hemichannels allow for the non-selective passage of molecules up to 1,000 Da.

3. Cx43 channels are permeable to both small molecules and macromolecules, makes them highly attractive targets for delivering drugs.

4. Since Cx43 is a connexin hemi channel and supports drug delivery, we infer that connexin hemi channels can be used for drug delivery.

**Step 1**

**Step Context:** connexin hemi channel are channels in cell membranes that can allow molecules to pass through and are part of a gap junction channels.

```
; =======================
; Schema
; =======================
; represents a molecule
(declare-sort Molecule)
; represents a channel
(declare-sort Channel)
; whether a channel is a connexin hemichannel
(declare-fun IsConnexinHemichannel (Channel) Bool)
; whether a channel is a gap junction channel
(declare-fun IsGapJunctionChannel (Channel) Bool)
; whether a channel is located in a cell membrane
(declare-fun IsInCellMembrane (Channel) Bool)
; whether an entity is considered a channel
(declare-fun IsChannel (Channel) Bool)
; whether a molecule can enter through a channel
(declare-fun CanEnterViaChannel (Molecule Channel) Bool)
; whether one channel is part of another channel
(declare-fun PartOf (Channel Channel) Bool)
; =======================
; Added premises
; =======================
; connexin hemmichannels are channels in membranes that pass molecules(
    documents)
(assert (forall ((c Channel))
  (=> (IsConnexinHemichannel c)
      (and (IsChannel c)
           (IsInCellMembrane c)
           (exists ((m Molecule)) (CanEnterViaChannel m c))))))

; Connexin hemichannels are part of gap junction channels ( documents)
(assert (forall ((h Channel))
  (=> (IsConnexinHemichannel h)
      (exists ((g Channel))
        (and (IsGapJunctionChannel g)
             (PartOf h g))))))

; =======================
; Step logic
; =======================
; Connexin hemichannels are channels in membranes that pass molecules
(assert (forall ((c Channel))
  (=> (IsConnexinHemichannel c)
      (and (IsChannel c)
           (IsInCellMembrane c)
           (exists ((m Molecule)) (CanEnterViaChannel m c))))))

; Connexin hemichannels are part of gap junction channels
(assert (forall ((h Channel))
  (=> (IsConnexinHemichannel h)
      (exists ((g Channel))
        (and (IsGapJunctionChannel g)
             (PartOf h g))))))
```

**Evaluation:** pass

Step 2

**Step Context:** Connexin hemi channels allow for the non-selective passage of molecules up to 1,000 Da.

```
; ======================
; Schema
; ======================
; returns the molecular weight of a molecule
(declare-fun MolecularWeight (Molecule) Real)
; ======================
; Added premises
; ======================
; Connexin hemmichannels allow for the non-selective passage of molecules up to
    1,000 Da.
(assert (forall ((m Molecule) (c Channel))
  (=> (and
        (IsConnexinHemichannel c)
        (CanEnterViaChannel m c))
      (<= (MolecularWeight m) 1000))))

; ======================
; Step logic
; ======================
; If a drug can enter via connexin channel, its weight must be <= 1000
(assert (forall ((m Molecule) (c Channel))
  (=> (and
        (IsConnexinHemichannel c)
        (CanEnterViaChannel m c))
      (<= (MolecularWeight m) 1000))))
```

**Evaluation:** Pass

Step 3

**Step Context:** Cx43 channels are permeable to both small molecules and macromolecules, makes them highly attractive targets for delivering drugs.

```
; ======================
; Schema
; ======================
; whether a channel is specifically connexin 43 (Cx43)
(declare-fun IsCx43 (Channel) Bool)
; whether a molecule is a small molecule
(declare-fun IsSmallMolecule (Molecule) Bool)
; whether a molecule is a macromolecule
(declare-fun IsMacromolecule (Molecule) Bool)
; whether a channel is permeable to a given molecule
(declare-fun PermeableTo (Channel Molecule) Bool)
; whether a channel can deliver drugs (channel-level, independent of molecule)
(declare-fun CanDeliverDrug (Channel) Bool)
; ======================
; Added premises
; ======================
; The permeability of Cx43 channels to small molecules and macromolecules, makes
;      them highly attractive targets for delivering drugs directly into the
;      cytoplasm. (document)
(assert
  (forall ((c Channel) (m Molecule))
    (=> (and (IsCx43 c)
             (or (IsSmallMolecule m)
                 (IsMacromolecule m)))
        (and (CanEnterViaChannel m c)
             (=> (CanEnterViaChannel m c)
                 (CanDeliverDrug c)))))))

; Any large macromolecule has a molecular weight > 1,000 Da. (commonsense)
(assert
  (forall ((m Molecule))
    (=> (IsMacromolecule m)
        (> (MolecularWeight m) 1000))))

; Any small macromolecule has a molecular weight <= 1,000 Da. (commonsense)
(assert
  (forall ((m Molecule))
    (=> (IsSmallMolecule m)
        (<= (MolecularWeight m) 1000))))

; ======================
; Step logic
; ======================
; Combined: if c is Cx43 and m is small or a macromolecule, then
; (1) m permeates via c; and (2) given that permeability, c can deliver drugs.
(assert
  (forall ((c Channel) (m Molecule))
    (=> (and (IsCx43 c)
             (or (IsSmallMolecule m)
                 (IsMacromolecule m)))
        (and (CanEnterViaChannel m c)
             (=> (CanEnterViaChannel m c)
                 (CanDeliverDrug c)))))))
```

**Evaluation:** Pass

Step 4

**Step Context:** Since Cx43 is a connexin hemi channel and supports drug delivery, we infer that connexin hemi channels can be used for drug delivery.

```
; ======================
; Schema
; ======================
; (none new)

; ======================
; Added premises
; ======================
; Cx43 channels are a type of connexin channel (Commonsense)
; Cx43 is a type of connexin hemichannel
(assert (forall ((c Channel))
    (=> (IsCx43 c)
        (IsConnexinHemichannel c))))

; Cx43 supports drug delivery.(Document)
(assert
    (exists ((c Channel))
        (and (IsCx43 c)
             (CanDeliverDrug c))))

; ======================
; Step logic
; ======================
;  Cx43 is a connexin hemi channel and supports drug delivery
(assert (forall ((c Channel))
    (=> (IsCx43 c)
        (IsConnexinHemichannel c))))

(assert
    (exists ((c Channel))
        (and (IsCx43 c)
             (CanDeliverDrug c))))

; Every drug that a Cx43 channel can deliver, can also be delivered by some
;     Connexin hemichannel.

(assert
    (forall ((c Channel))
        (=> (and (IsCx43 c)
                 (CanDeliverDrug c))
            (exists ((h Channel))
                (and (IsConnexinHemichannel h)
                     (CanDeliverDrug h))))))
```

**Evaluation:** Pass

### A.4.3 UNTRANSLATABLE EXAMPLE FROM SARA

Untranslatable example from SARA

**Reference information:** Section 3306(c)(1) applies to agricultural labor.

**Question:** Determine whether the following statements are entailed under the statute. Description: Alice has paid $3200 to Bob for agricultural labor done from Feb 1st, 2017 to Sep 2nd, 2017. Alice is an American employer. Statement: Section 3306(c)(1) applies to Alice employing Bob for the year 2017. Reply with either: Entailment, Contradiction.

**Answer:** Entailment

**Original CoT:**
Let me solve this step by step.

First, let's identify the key elements in the statute:
* The statute in question is Section 3306(c)(1).
* Its primary subject is "agricultural labor".
Let's analyze the given description of the situation:
* Employee: Bob
* Employer: Alice (an American employer)
* Type of work: The work is explicitly identified as agricultural labor.
Let's compare the situation to the statute's basic scope:
* The statute, Section 3306(c)(1), specifically and directly addresses agricultural labor.
* The work performed by Bob is agricultural labor.

> \* Therefore, the situation falls squarely within the fundamental subject matter of this statutory section.
> Important consideration:
> \* The fact that the work was performed for an American employer raises potential jurisdictional or geographical questions.
> \* However, the provided statute fragment does not contain any information about geographical limitations.
> Based on the basic provision:
> \* Since the work is agricultural labor, and Section 3306(c)(1) explicitly covers agricultural labor, the situation described falls under the basic scope of this section.

VERICoT Output before self-reflection:

**Decomposed CoT Steps**

1. Section 3306(c)(1) of the statute addresses agricultural labor
2. The work performed by Bob for Alice is specifically agricultural labor
3. The basic scope of Section 3306(c)(1) covers all agricultural labor
4. While Alice is an American employer and there might be jurisdictional implications, the statute fragment does not provide information about geographical limitations
5. Bob's employment falls under the basic scope of Section 3306(c)(1)

**Step 1**

**Step Context:** Section 3306(c)(1) of the statute addresses agricultural labor

```
; =======================
; Schema:
; =======================
; represents a section of statute
(declare-sort StatuteSection)

; specific section 3306(c)(1)
(declare-const section_3306c1 StatuteSection)
; whether section deals with agricultural labor
(declare-fun deals_with_agricultural_labor (StatuteSection) Bool)

; =======================
; Add premises:
; =======================
; Section 3306(c)(1) applies to agricultural labor (document)
(assert
  (deals_with_agricultural_labor section_3306c1))

; =======================
; Step Logic:
; =======================
; sec 3306c1 abt agri labor
(assert
  (deals_with_agricultural_labor section_3306c1))
```

**Evaluation:** Pass

**Step 2**

**Step Context:** The work performed by Bob for Alice is specifically agricultural labor

```
; ======================
; Add schema:
; ======================
; represents a person
(declare-sort Person)
; represents Alice
(declare-const alice Person)
; represents Bob
(declare-const bob Person)
; whether person performs agricultural labor
(declare-fun performs_agricultural_labor (Person) Bool)
; whether person employs another person
(declare-fun employs (Person Person) Bool)

; ======================
; Add premises:
; ======================
; Alice employed Bob to perform agricultural labor from February 1st to
;     September 2nd, 2017, as evidenced by Alice's payment of $3200 for this work
;       (document)
; Dates encoded as YYYYMMDD
; paid employer worker laborType amount startDate endDate
(declare-fun paid (Person Person LaborType Int Int Int) Bool)

(assert
  (and
    (employs alice bob)
    (performs_agricultural_labor bob)
    (paid alice bob agricultural_labor 3200 20170201 20170902)))

; ======================
; Step logic:
; ======================
; Bob performs agri labor for Alice
(assert
  (and
    (performs_agricultural_labor bob)
    (employs alice bob)))
```

**Evaluation:** Pass

---

**Step 3**

**Step Context:** The basic scope of Section 3306(c)(1) covers all agricultural labor

```
; ======================
; Add schema:
; ======================
; predicate: whether a person's work is covered by 3306(c)(1)
(declare-fun covered_by_section_3306c1 (Person) Bool)

; ======================
; Add premises:
; ======================
; A person performs labor, therefore anyone perform agricultural labor is
;       covered by Section 3306(c)(1) (commonsense)
(assert
  (forall ((p Person))
    (=> (performs_agricultural_labor p)
        (covered_by_section_3306c1 p))))

; ======================
; Step logic:
; ======================
; sec 3306c1 covers agri labor
; if person does agri labor then covered by sec 3306c1
(assert
  (deals_with_agricultural_labor section_3306c1))
(assert
  (forall ((p Person))
    (=> (performs_agricultural_labor p)
        (covered_by_section_3306c1 p))))
```

**Evaluation:** Pass

---

Step 4

**Step Context:** While Alice is an American employer and there might be jurisdictional implications, the statute fragment does not provide information about geographical limitations

```
; ======================
; Schema:
; (none new)
; ======================

; ======================
; Add premises:
; (none new)
; ======================

; ======================
; Step logic:
; (none new)
; ======================
```

**Evaluation:** Fail due to Untranslatable
*Reason: The phrase "there might be jurisdictional implication" cannot be rendered under SMI-LIB rules because it expresses a possibility rather than a definitive statement."*

---

Step 5

**Step Context:** Bob's employment falls under the basic scope of Section 3306(c)(1)

```
; =======================
; Schema:
; (none new)
; =======================

; =======================
; Add premises:
; (none new)
; =======================

; =======================
; Step logic:
; =======================
(assert  (covered_by_section_3306c1 bob))
```

**Evaluation:** Pass

VERICOT Output after self-reflection:

**Decomposed CoT steps**

1. Section 3306(c)(1) explicitly applied to agricultural labor as shown by the statute text 'agricultural labor'.
2. Bob performed agricultural labor.
3. Since Bob's work qualifies as agricultural labor, and Section 3306(c)(1) covers agricultural labor, Bob's employment falls under the basic scope of Section 3306(c)(1).

**Step 1**

**Step Context:** Section 3306(c)(1) explicitly addresses agricultural labor, which means that any person who performs agricultural labor is covered by Section 3306(c)(1).

```
; =======================
; Schema
; =======================
; represents a statute section
(declare-sort StatuteSection)
; represents a type of labor
(declare-sort LaborType)
; represents a person
(declare-sort Person)
; specific statute section 3306(c)(1)
(declare-const section_3306c1 StatuteSection)
; specific labor type: agricultural labor
(declare-const labor_agricultural LaborType)
; whether a section addresses a labor type
(declare-fun section_addresses_labor (StatuteSection LaborType) Bool)
; whether a person performed a labor type
(declare-fun performed_labor (Person LaborType) Bool)
; whether a person's employment is covered by a statute section
(declare-fun employment_covered (Person StatuteSection) Bool)

; =======================
; Added premises
; =======================
; Section 3306(c)(1) explicitly applies to agricultural labor as shown by the
;     statute text 'agricultural labor' (document)
(assert (section_addresses_labor section_3306c1 labor_agricultural))

; A person performs labor, therefore anyone performing agricultural labor is
;     covered by Section 3306(c)(1) (commonsense)
(assert
  (forall ((p Person))
    (=> (performed_labor p labor_agricultural)
        (employment_covered p section_3306c1))))

; =======================
; Step logic
; =======================
; Section 3306(c)(1) explicitly addresses agricultural labor
; -> anyone who performed agricultural labor is covered
(assert (section_addresses_labor section_3306c1 labor_agricultural))
(assert
  (forall ((p Person))
    (=> (performed_labor p labor_agricultural)
        (employment_covered p section_3306c1))))
```

**Evaluation:** Pass

Step 2

**Step Context:** Bob performed agricultural labor.

```
; =======================
; Schema
; =======================
; Bob performed agricultural labor.
(declare-const Bob   Person)

; =======================
; Added premises
; =======================
; Alice has paid $3200 to Bob for agricultural labor done from Feb 1st, 2017 to
    Sep 2nd, 2017. (document)
; Dates encoded as YYYYMMDD
; paid employer worker laborType amount startDate endDate
(declare-fun paid (Person Person LaborType Int Int Int) Bool)
(declare-const Alice Person)
(assert (performed_labor Bob labor_agricultural))
(assert (paid Alice Bob labor_agricultural 3200 20170201 20170902))

; =======================
; Step logic
; =======================
(assert (performed_labor Bob labor_agricultural))
```

**Evaluation:** Pass

Step 3

**Step Context:** Since Bob's work qualifies as agricultural labor, and Section 3306(c)(1) covers agricultural labor, Bob's employment falls under the basic scope of Section 3306(c)(1).

```
; =======================
; Schema
; (none new)
; =======================

; =======================
; Added premises
; (none new)
; =======================

; =======================
; Step logic
; =======================
; derive Bob's coverage
(assert (=>
  (and
    (performed_labor Bob labor_agricultural)
    (forall ((p Person))
      (=> (performed_labor p labor_agricultural)
          (employment_covered p section_3306c1))))
  (employment_covered Bob section_3306c1)))
```

**Evaluation:** Pass