# OpenReview forum: "VeriCoT: Neuro-symbolic Chain-of-Thought Validation via Logical Consistency Checks"
_ICLR.cc/2026/Conference — ICLR 2026 Poster_

### Official Review · Reviewer_37xu · 2025-10-25

**Soundness:** 2
**Presentation:** 4
**Contribution:** 3
**Rating:** 4
**Confidence:** 4

**Summary:**

This paper introduces VERICOT, a neuro-symbolic method designed to verify reasoning trace of LLMs.
The methodology consists of two main components: 1. Autoformalization – translating each natural-language CoT step into a precise formal logic representation and 2. Verification – using automated constraint solvers to check whether the formal statement is logically consistent and entailed by explicitly identified premises. VERICOT demonstrates effectiveness across diverse tasks (ProofWriter, LegalBench, BioASQ), reliably detecting logical errors. Also, The authors also apply this methodology in two use cases—real-time self-reflection and constructing high-fidelity datasets—to actively improve LLM reasoning quality and task performance.

**Strengths:**

1. This paper deals with relatively unfamiliar concepts such as neuro-symbolic reasoning and first-order logic, yet it is written in a way that is easy for readers to follow. The authors also provide appropriate examples that effectively aid understanding.

2. The overall approach of the paper is well thought out. Decomposing the CoT into atomic units, expressing them in a formal language, and verifying them is a promising direction for improving the interpretability of reasoning.

3. The categorization presented in the paper is clear and logically sound.

4. The two-step problem-solving process—formalization and premise generation—is well designed, and the authors effectively demonstrate the usefulness of their method through its applications in real-time inference and dataset generation.

5. The paper makes appropriate use of the LLM-as-a-judge, relying on it only partially rather than excessively, and successfully leveraging its advantages.

**Weaknesses:**

1. Questions about the metrics presented in Table 1

1-1. I have several questions regarding the metrics used in the experimental results shown in Table 1.
The Pass Rate measures whether a CoT trace is verifiable. Did each baseline method (ER or DSB) apply its own verification method, different from VeriCOT’s?
If so, I’m not sure a higher Pass Rate necessarily indicates a better verification method. Some verification methods can be very strict and reject even minor leaps in reasoning, while others may be more lenient and mark more traces as “pass.”
Wouldn’t it be better to apply a stricter verification method, along with the self-reflection mechanism shown in Section 3.4, to produce higher-quality CoTs?

1-2. Related to the above, as I understand it, VCAR represents the proportion of test cases that satisfy both Pass Rate and Precision. If Pass Rate is high, then VCAR will naturally appear favorable. However, as mentioned in 1-1, if a high Pass Rate doesn’t necessarily mean a better verification process, doesn’t that also undermine the interpretive value of the VCAR metric?

1-3. In my opinion, among all the metrics for CoT verification, Precision is the most important.
This is because we examine LLMs’ CoTs under the expectation that if their reasoning is logical and sound (even when we don’t know the ground truth), their final answer is likely correct.
Therefore, Precision—the proportion of truly correct CoTs among those verified by a given method—seems the most meaningful and task-relevant metric.
However, according to Table 1, VeriCOT does not achieve higher Precision than the baselines, and in the case of ProofWriter, its Precision is even lower than VeriCOT-NoPrem. The authors should provide a justification for this.

1-4. What is the reason for the differences in Task Accuracy among methods? Were all methods evaluated on the same CoT samples?

2. Since the proposed approach verifies CoTs through multiple rounds of LLM inference, it would be helpful to report the average verification cost per CoT (e.g., number of LLM calls, computation cost, etc.).

3. Premise generation and premise evaluation—two core components of the proposed method—both rely on LLMs. Consequently, performance may vary greatly depending on which LLM is used. However, the experiments were conducted solely with Claude 3.5 Sonnet.
It seems essential to analyze how performance changes across LLMs with different capabilities.
If results vary by model, the authors should explain why; if they do not, an explanation for the stability would also be valuable.

4. It would be helpful to include qualitative examples of the Commonsense type premises among the three types of NL Premises.
The classification in Table 2—Grounded contextual premise, Acceptable commonsense premise, and Necessary commonsense premise—is somewhat difficult to evaluate in terms of reliability.
What exactly distinguishes something “acceptable” from “necessary”?
A clearer analysis of what counts as a commonsense premise (and what does not) would strengthen the paper.
Since the proposed method’s Pass Rate likely depends heavily on how well the model generates commonsense premises, this aspect deserves more attention.

5. It seems that this paper assumes the CoT is already divided into individual steps, but an explanation of how the CoT steps were segmented would be necessary.


Overall, the paper’s motivation, proposed solution, structure, and writing quality are excellent. However, I believe the experimental results are still insufficient to convincingly demonstrate the practical usefulness of the proposed method.

**Questions:**

The main questions are in the Weaknesses section.

Additionally, (This is not a drawback, but just a question:)
Could this methodology also be applied to reasoning traces produced by reasoning-scaled models trained with reinforcement learning (RL), rather than CoTs generated purely through prompting?

---

> ### Author Response · Authors · 2025-11-21
>
> We sincerely thank reviewer 37xu for the valuable comments. Here we answer all your questions.
>
> **1-1.2.3 Questions about the metrics presented in Table 1**
>
> We appreciate the critique that a high pass rate might imply leniency. However, VeriCoT achieves this rate not through leniency, but through a highly structured task decomposition that leverages both the formal solver's rigor and the LLM's capabilities. The baselines’ low pass rates are not due to stricter evaluation, but rather of limitations in their evaluation mechanisms, which cause them to fail on all but only a small number of trivial cases.
>
> While we value high precision, precision must be understood in light of the precision–recall trade-off: accepting more cases (higher pass rate/recall) naturally lowers precision. We observe that  the baselines achieve high precision on ProofWriter by "cherry-picking" the easiest cases and abstaining on the rest (resulting in very low pass rates). We strongly agree with the reviewer that Precision is a first-class citizen in this evaluation; however, when pass rate is extremely low compared to our method, it hinders usefulness. We agree as well that “a high pass rate doesn’t necessarily mean a better verification process” – however, given VCAR is the product of pass rate and precision, we posit that it does provide interpretative value that incorporates both the leniency and accuracy of the verifiers. The metric penalizes both the verifier that is too lenient (high pass rate, low precision) and the verifier that is overly selective and only accepts trivial cases (low pass rate, high precision).
>
> **1-4. Differences in task accuracy among methods**
>
> In DSB (Direct SMT Baseline), the model decomposes the original CoT into its own CoT steps while simultaneously translating them into formalizations. This joint generation can introduce noise and cause the decomposed steps to differ from the original CoT. For a fair comparison, we report DSB’s task accuracy using these newly generated decomposed steps. All other baselines are evaluated on the same shared CoT samples.
>
> **2. Average verification cost per CoT**
>
> On average, each example requires 45 total LLM calls across all datasets, with some parts of the algorithm running in parallel, for example, logic translation for multiple premises in a step.
>
> **3. LLM performance on premise generation and premise evaluation**
>
> We conducted  a human analysis by sampling 20 examples per dataset, evenly across the four labels: verifiable, ungrounded, untranslatable, and contradiction. We have updated the draft to reflect this. We assess whether each logic translation is accurate, faithfully captures all necessary details, and preserves the correct logical relationships. We also evaluate the quality of the generated premises: (1) whether a contextual premise is correctly grounded in the source document or question, and (2) whether a commonsense premise is acceptable and reasonable given the context and the targeted reasoning step. Finally, we check whether LLMaaj’s judgments align with human annotations. Across all criteria, we observe consistently high performance. Notably, LLMaaJ decisions on premises are aligned with human judgements 95-100% of the time.
> | Dataset      | Grounded contextual premises / all premises | Acceptable commonsense premises / all premises | Correct LLMaaJ / all premises |
> |--------------|----------------------------------------|--------------------------------------------|-----------------------------------------------|
> | BioASQ      | 0.87                                       | 0.91                                          | 0.97                           |
> | SARA         | 0.92                                       | 0.89                                          | 0.95                           |
> | Proofwriter  | 0.95                                       | 0.95                                          | 1.00                           |
>
> We ran the same test on openai/gpt-oss-120b using the same setup and found that its premise generation and premise evaluation quality were comparable to Claude-3.5-Sonnet-V2 in our human analysis.
>
> | Dataset      | Grounded contextual premises / all premises | Acceptable commonsense premises / all premises | Correct LLMaaJ / all premises |
> |--------------|----------------------------------------|--------------------------------------------|-----------------------------------------------|
> | BioASQ      | 0.85                                       | 0.86                                          | 0.93                           |
> | SARA         | 0.91                                       | 0.91                                          | 0.92                           |
> | Proofwriter  | 0.93                                       | 0.94                                          | 0.96                           |
>
> We thus conclude that LLM usage for premise-related atomic tasks can be handled by reasonably capable models.

---

> > ### Author Response · Authors · 2025-11-21
> >
> > **4. Qualitative examples of the commonsense type premises**
> >
> > Here are two qualitative examples with commonsense-type premises. We note that ‘acceptable’ and ‘necessary’ are not mutually inclusive. We will accept a commonsense premise if it is reasonable to a typical human given the context and the targeted reasoning step. Separately, we report whether a commonsense premise is necessary as a signal for future improvement, since ideally we would like to reduce the number of acceptable but unnecessary premises. We have updated the draft to reflect this.
> >
> > Example 1 from SARA:
> > ```
> > Document: … Statute  section 1(a)(i) 15% of taxable income if the taxable income is not over $36,900; …
> > Question: Alice's taxable income for the year 2017 is $25561. How much taxes does Alice have to pay for the year 2017 under Statute section 1(a)(i)?
> >
> >
> > CoT:
> > Step 1: The statute specifies a tax rate of 15% for taxable income not exceeding $36,900
> > Step 2: Alice's taxable income for 2017 is $25,561
> > Step 3: Since $25,561 is less than $36,900, Alice's income falls within this tax bracket
> > Step 4: The tax calculation for Alice's income of $25,561 at 15% equals $3,834.15, which rounds to $3,834
> > Step 5: Alice must pay $3,834 in taxes
> >
> > Final generated  premises:
> > Document:
> > 1. The tax rate is exactly 15% for taxable income not over $36,900
> > Question:
> > 1. Alice's taxable income for the year 2017 is $25,561
> > Commonsense:
> > 1. The tax amount is calculated by multiplying the taxable income by the tax rate percentage divided by 100
> > 2. The rounded tax amount is the exact tax amount rounded to the nearest whole dollar, where amounts with cents less than 50 round down and amounts with 50 cents or more round up.
> > ```
> > In this example all contextual premises are grounded and all commonsense premises are acceptable and necessary.
> >
> > Example 2 from Poofwriter:
> > ```
> > Reference information: Bob is big. Bob is nice. Bob is smart. Charlie is nice. Dave is nice. Erin is big. Erin is blue. Erin is furry. Erin is quiet. Erin is round. Erin is smart. All nice things are big. All furry things are blue. Blue, furry things are quiet. If something is round and nice then it is blue. If Bob is round then Bob is blue. Big, quiet things are furry. If something is blue and furry then it is smart. If Charlie is nice and Charlie is big then Charlie is quiet.
> >
> > Question: Based on the above information, is the following statement true, false, or unknown? Charlie is blue.
> >
> > CoT:
> > Step 1: Charlie is nice
> > Step 2: Since all nice things are big, and Charlie is nice, Charlie must be big
> > Step 3: Since Charlie is both nice and big, and there is a rule stating 'If Charlie is nice and Charlie is big then Charlie is quiet', Charlie must be quiet
> > Step 4: Since Charlie is both big and quiet, and there is a rule stating 'Big, quiet things are furry', Charlie must be furry
> > Step 5: Since Charlie is furry, and there is a rule stating 'All furry things are blue', Charlie must be blue
> > Step 6: Therefore, the statement 'Charlie is blue' is true
> >
> > Final generated premises:
> > Document:
> > 1. Charlie is nice
> > 2. All nice things are big
> > 3. If Charlie is nice and Charlie is big then Charlie is quiet
> > 4. Big, quiet things are furry
> > 5. All furry things are blue
> > Commonsense:
> > 1. Being big is a property that some things have and others don't
> > 2. Being quiet is a property that some things have and others don't
> > ```
> > In this example all contextual premises are grounded and all commonsense premises are acceptable but unnecessary. We have updated the draft to reflect this.
> >
> > **5. CoT step decomposition**
> >
> > We explicitly prompt the model to decompose the CoT into individual steps. A human analysis of 60 examples (20 per dataset) showed that in 96.7% of instances, the decomposition is accurate to faithfully preserved information from the original CoT. We have updated the draft to reflect this.
> >
> > **Q1. Applicability of VeriCoT to RL-trained LLM reasoning traces**
> >
> > Yes! VeriCoT can be used to analyze these traces in reasoning-scaled models, but it should not be used as the evaluation metric because the reasoning traces naturally contain exploration, backtracking, and contradictions. However, our method can definitely be applied to evaluate the final reasoning trajectory that leads to the final answer in these reasoning traces.

---

### Official Review · Reviewer_MriA · 2025-10-27

**Soundness:** 3
**Presentation:** 3
**Contribution:** 3
**Rating:** 6
**Confidence:** 4

**Summary:**

This paper augments CoT with generating verifiable first-order logic expressions, which can be used by a constraint solver, to verify the chains of thought being generated by LLMs. The paper spells out a careful procedure to do this (Section 2), which consists of generating a formal logical expression from each COT step ($C_i \rightarrow F_i$), validating the formal expression, then generating premises to support the formal expression ($F_i \rightarrow P_i$).

This procedure is especially nice in characterizing the unreliable CoT steps (ex: contradictions, ungrounded reasoning, etc.).

The authors then demonstrate the benefits of their procedure on a few benchmarks that rely on reasoning (ProofWriter, LegalBench, BioASQ). Augmenting VeriCOT does improve the reliability of the model's CoT steps (as measured by precision, pass rate), but does not seem to affect the end task accuracy much.

The authors study the effect of fine-tuning LLMs with VeriCOT. While simply applying SFT does not seem to make much of an effect, applying DPO (to favor correct VeriCOT steps vs. incorrect VeriCOT steps) does improve task accuracy, suggesting a better post-training recipe for domains that require multiple logical reasoning steps.

**Strengths:**

This paper provides an intuitive procedure to validate the reliability of CoT sequences. In particular, looking at the VeriCOT failure cases is a good way of uncovering common "fallacies" of model CoTs (e.g., contradictions, ungrounded reasoning).

Overall, the paper makes a nice coherent, and well-scoped contribution to test the reliability of LLM's CoT steps.

**Weaknesses:**

From the brief description of the baseline methods being compared to, it sounds like VeriCOT isn't the first to incorporate FOL (or similar expressions that is conducive to do verification along the way) to validate and improve CoT. It would be helpful to explicitly spell out how VeriCOT differs from these other approaches.

Should there be another baseline where the LLM-judge just verifies each natural language reasoning step (without the FOL counterparts) to isolate the effect of FOL steps?

While VeriCOT seems to improve on precision and pass rate, it doesn't seem to affect task accuracy that much. Does this suggest that it's not so much the correctness of the CoT that matters, but instead something else like generation diversity? What are your thoughts on why this is?

VeriCOT relies heavily on a LLM to correctly judge the generated premises as valid or invalid. How reliable are Claude's judgements for this?

**Questions:**

What is the difference between updating $P_i$ with $P_i = P_{i-1}$ vs. $P_i = P_{i-1} \cup \{P_i\}$?

In Table 4 - it doesn't seem like SFT-ing with VeriCOT makes much of a difference, unless DPO is applied. Any thoughts as to why?

How does VCAR combine precision and pass rate? It seems a bit odd to combine the two given that the two measures seem to have very different distributions.

Why do you think precision drops after self-reflection for ProofWriter?

What results do you expect to see when VeriCOT is applied to "unfaithful" CoT, in which models justify their answers "post-hoc"? (https://arxiv.org/abs/2305.04388)

---

> ### Author Response · Authors · 2025-11-21
>
> We sincerely thank reviewer MriA for the valuable comments. Here we answer all your questions.
>
> **W1. How VeriCoT differs from prior FOL-based CoT verification approaches**
>
> As noted in the related work on solver-based verification of NL reasoning (Lines 471–477),  the most  closely related work is Quan et al. (ER, https://aclanthology.org/2024.emnlp-main.172.pdf ), but it works on single-step explanations for pure entailment tasks and does not address grounding the reasoning or multi-step evaluation.
>
> **W2. Pure NL LLM-judge baseline**
>
> We appreciate the suggestion of adding a baseline where an LLM judge verifies each natural-language reasoning step without any FoL formalization.  While this could be an interesting direction, in our work, we focus on FoL because it enables formal verification and guaranteed soundness, advantages that NL-only judging cannot provide. We considered only methods that mirrored our desired I/O: as input a CoT, and as output, a sound logic formula and NL underlying premise list. A NL-only baseline will require a completely different set of evaluation rubrics and judging criteria, making it not directly comparable to our FoL-based setup.
>
> **W3. Why VeriCoT Improves CoT Precision but Not Overall Task Accuracy**
>
> Our goal is to build LLM reasoning tools that are more verifiable using interpretable symbolic logic. Given two approaches with the same task accuracy but different levels of verifiability, we strongly prefer the one that is verifiable.
>
> We therefore want to encourage LLMs, through inference-time verification or fine-tuning, to be more likely to generate CoTs that are verifiable through our method. Specifically we encourage CoTs that can be autoformalized into a clean FoL reasoning trace with an interpretable set of underlying atomic premises. We want to _discourage_ the LLM from generating CoTs that cannot be verified in this way, especially if a verifiable alternative could have been generated.
>
> We thus view CoT validity and final-answer accuracy as related but ultimately separate problems. VeriCoT primarily improves the soundness of the CoT, which raises precision and pass rate. But final accuracy also depends on factors beyond reasoning correctness, such as model prior knowledge and search diversity.
>
> **W4. Reliability of LLM-Based Premise Judgments in VeriCoT**
>
> We conduct a human analysis by sampling 20 examples per dataset, evenly across the four labels: verifiable, ungrounded, untranslatable, and contradiction. We manually evaluate the correctness of LLMaaj’s judgments. We observe that LLMaaJ decisions on premises are aligned with human judgements 95-100% of the time.
> | Dataset      | Correct LLMaaJ / all premises |
> |--------------|----------------------------------------|
> | BioASQ      | 0.97                           |
> | SARA         | 0.95                           |
> | Proofwriter  | 1.00                           |
>
> **Q1. What is the difference between updating $P_i$  with  $P_i =P_{i-1}$  vs.  $P_i = P_{i-1} \cup P_i$ ?**
>
> If the current step can pass the checks without introducing new premises, $P_i$ is updated with $P_i =P_{i-1}$, meaning the current step can be entailed by previous steps. If it does not, we generate the necessary new premises and update the set accordingly to include them: $P_i = P_{i-1} \cup P_i$.
>
> **Q2. In Table 4 - it doesn't seem like SFT-ing with VeriCOT makes much of a difference, unless DPO is applied. Any thoughts as to why?**
>
> As noted in Section 3.4 (lines 455–458), SFT yields an average 3% improvement in final accuracy because the instances that pass verification tend to have higher precision than the overall task accuracy. However, SFT does not provide explicit guidance about what kinds of CoT are preferred or more verifiable. In contrast, DPO offers explicit preference signals, making it more effective for encouraging CoT that can be successfully verified.
>
> **Q3. How does VCAR combine precision and pass rate? It seems a bit odd to combine the two given that the two measures seem to have very different distributions.**
>
> VCAR = precision * pass rate, which is the percentage of the instances which are both verifiable and correct. This is somewhat analogous to the relationship between F1 score and precision/recall. Here pass rate is how often CoTs are deemed verifiable and precision is the proportion of correct answers among verifiable CoTs. VCAR is designed to balance the two competing objectives. The metric penalizes both the verifier that is too lenient (high pass rate, low precision) and the verifier that is overly selective and only accepts trivial cases (low pass rate, high precision). VCAR better captures verification quality to the extent that ground-truth answer labels are correlated with correct reasoning.

---

> ### Author Response · Authors · 2025-11-21
>
> **Q4. Why do you think precision drops after self-reflection for ProofWriter?**
>
> We conducted a qualitative analysis and did not observe a clear pattern in the failure cases, as they are very rare. One possible explanation is that the pre–self-reflection CoT may occasionally rely on guessing, whereas after self-reflection the model becomes more inclined to quote from the document, which can cause it to become more focused on extracting relevant facts from the document and may lose track of the underlying logic and ultimately produce a wrong ‘unknown’ answer. We present an example as below:
>
> ```
> Reference information: The cat eats the squirrel. The cat sees the squirrel. The cow eats the squirrel. The cow sees the cat. The rabbit is round. The rabbit sees the cat. The squirrel eats the rabbit. The squirrel is cold. The squirrel needs the rabbit. The squirrel sees the cat. If someone sees the cat and they are not green then they see the cow. If the rabbit is kind and the rabbit sees the squirrel then the squirrel needs the rabbit. Rough people are cold. If someone sees the rabbit then they are not round. If someone sees the squirrel and they are not green then they need the squirrel. If someone eats the cow then they see the rabbit. If someone eats the squirrel then they are rough. If someone is cold then they eat the cow.
>
> Question: Based on the above information, is the following statement true, false, or unknown? The cat is not round.
>
> CoT before self-reflection:
> Step 1: The task requires determining if there is sufficient information to conclude whether the cat is round or not round
> Step 3: The document states that anyone who is cold eats the cow, and anyone who eats the cow sees the rabbit.
> Step 4: In the rules, anything that sees the rabbit is explicitly said to be not round.
> Step 5: Since the cat fits into the conditions that lead to seeing the rabbit, the cat falls under the rule that makes it not round.
> Step 6: Therefore, the statement “The cat is not round” is true.
>
>
> CoT after self-reflection:
> Step 1: The task requires determining whether the given rules provide enough information to conclude that the cat is not round.
> Step 2: The document states that if someone eats the squirrel, then they are rough.
> Step 3: Rough people are cold, and if someone is cold then they eat the cow.
> Step 4: If someone eats the cow then they see the rabbit.
> Step 5: According to the rule, if someone sees the rabbit, then they are not round.
> Step 6: Since there is insufficient information to determine who sees the rabbit, the statement 'The cat is not round' must be classified as unknown.
> ```
>
> **Q5. What results do you expect to see when VeriCOT is applied to "unfaithful" CoT, in which models justify their answers "post-hoc"? (https://arxiv.org/abs/2305.04388)**
>
> Thank you for the interesting and thoughtful question! VeriCoT’s evaluation focuses on the logical soundness of the final, presented reasoning trace, rather than the model’s internal cognitive process. If a post-hoc or ‘unfaithful’ CoT still provides a logically valid and formally verifiable argument supporting the final answer, VeriCoT will mark it as verifiable. For many practical settings, it is sufficient that the model can present a clear, checkable CoT, even if that CoT does not reflect its internal generation path. VeriCoT is therefore designed to assess verifiable soundness, not to recover or interpret the model’s internal reasoning process.

---

### Official Review · Reviewer_D7eV · 2025-10-29

**Soundness:** 3
**Presentation:** 3
**Contribution:** 3
**Rating:** 8
**Confidence:** 4

**Summary:**

The paper describes a system that provides a multi-step process for transforming a Chain of Thought into a formal argument, with the goals of grounding and validating individual steps in the CoT and identifying logical errors in the CoT. This process answers three questions: can the CoT admit of auto-formalization, are the steps in the CoT supported, and what premisses does the CoT rely upon. This approach accomplishes two objectives: inference-time checking of the correctness of a CoT, and providing examples of CoTs that are verified to be correct for use in the fine-tuning of a reasoning model.

**Strengths:**

The core innovation lies in combining LLM-based autoformalization with SMT solver verification using Z3, which provides formal guarantees about logical consistency within the formalized representation. This approach offers more rigor than purely LLM-based verification methods, while remaining more flexible than pure formal methods that require pre-specified logical frameworks.

I see an analogy with the informal process of proof checking of a human-generated proof in informal and formal reasoning. The idea of using this to validate CoTs, and then using the validated CoT through fine tuning in a self-improving process is interesting. This concept
demonstrates practical versatility, and suggests a path toward increasingly reliable reasoning systems.

**Weaknesses:**

A fundamental concern is the circular dependency created by using LLMs for both autoformalization and premise generation, which are then employed to verify LLM outputs. This architectural decision introduces potential for systematic biases where the verifier may share failure modes with the system being verified. While the paper acknowledges this limitation, it doesn't adequately explore the implications or propose mitigation strategies. The system essentially asks LLMs to judge their own reasoning through a different lens, which may not catch errors that stem from fundamental misunderstandings encoded in the model family.

The presence of an "untranslatable" error category reveals a significant limitation in the approach's scope. Certain reasoning steps fundamentally cannot be represented in the supported first-order logic fragment, as illustrated by the LegalBench example where probabilistic statements like "might be jurisdictional implications" cannot be encoded. The paper reports these untranslatable rates but offers no strategies for handling such cases beyond marking them as failures, which limits the system's applicability to real-world scenarios where uncertainty or non-classical reasoning may be needed.

The reliance on LLM-as-judge evaluation for premise quality assessment introduces another layer of potential error and bias. While Table 2 reports high acceptance rates between 77% and 93%, these metrics don't guarantee that premises are actually correct or complete. The circular dependency problem resurfaces here, as LLMs evaluate premises generated by LLMs, potentially missing systematic errors or hallucinations.

**Questions:**

What specific technical advances would be required to achieve verification rates sufficient for practical use? Would a human-in-the-loop approach for handling untranslatable cases be viable?

---

> ### Author Response · Authors · 2025-11-21
>
> We sincerely thank reviewer D7eV for the constructive suggestions.  Here we clarify the questions you posed.
>
> **W. Systemic risks of circular LLM dependencies**
>
> We fully appreciate the reviewer’s point, and we agree that some systematic bias in LLM usage is naturally unavoidable. Our goal is to design a structured task decomposition that reduces these biases by combining the strengths of formal solvers and LLMs. The SMT solver handles all formal reasoning, while the LLM is only asked to perform simple translation under a well-designed algorithm, generate premises from well-defined context, and small checks on isolated premises. Prior work (e.g., FactScore, https://aclanthology.org/2023.emnlp-main.741.pdf ) shows that LLMs are much more reliable on such atomic judgments than on evaluating long chains of reasoning. This decomposition avoids compounding errors and also surfaces clear, human-readable premises, enabling targeted and transparent human review when needed. While FoL does come with representation limits, acknowledged in our limitations section, we believe it remains a valuable direction, as a meaningful portion of CoT traces in the domains we consider can be formalized and benefit from the rigor of solver-based verification.
>
> We also conduct a human analysis by sampling 20 examples per dataset, evenly across the four labels: verifiable, ungrounded, untranslatable, and contradiction. We assess whether each logic translation is accurate, faithfully captures all necessary details, and preserves the correct logical relationships. We also evaluate the quality of the generated premises: (1) whether a contextual premise is correctly grounded in the source document or question, and (2) whether a commonsense premise is acceptable and reasonable given the context and the targeted reasoning step. Finally, we check whether LLMaaj’s judgments align with human annotations. Across all criteria, we observe consistently high performance. The autoformalizer exhibits a relatively low error rate, under 20%, which is already quite strong for NL-to-FoL translation. LLMaaJ decisions on premises are aligned with human judgements 95-100% of the time.
> | Dataset      | Accurate translation / all statements | Grounded contextual premises / all premises | Acceptable commonsense premises / all premises | Correct LLMaaJ / all premises |
> |--------------|----------------------------------------|--------------------------------------------|-----------------------------------------------|--------------------------------|
> | BioASQ       | 0.80                                   | 0.87                                       | 0.91                                          | 0.97                           |
> | SARA         | 0.86                                   | 0.92                                       | 0.89                                          | 0.95                           |
> | Proofwriter  | 0.87                                   | 0.95                                       | 0.95                                          | 1.00                           |
>
> **Q. Possible technical advances to achieve verification rates sufficient for practical use**
>
> Thank you for the thoughtful question! We could encourage more targeted self-repair at different stages, such as during autoformalization or full CoT-step repair. A human-in-the-loop could also 1) assist with portions of the CoT that cannot be translated by extracting key information or rewriting it so that it becomes expressible in FoL; 2) we could collect translation failures, consult formal-methods experts, and adopt more expressive logical formalisms to improve translation coverage.

---

### Official Review · Reviewer_C6Nf · 2025-11-01

**Soundness:** 3
**Presentation:** 3
**Contribution:** 3
**Rating:** 2
**Confidence:** 5

**Summary:**

This paper introduces VERICOT, a neuro-symbolic framework to validate the logical consistency of Chain-of-Thought (CoT) reasoning. It autoformalizes each CoT step into First-Order Logic (FOL) and uses an SMT solver (Z3) for verification. The authors claim their core novelty is a "premise generation" step, where an LLM infers and formalizes implicit commonsense or contextual premises to complete logical gaps. The resulting verification signal is then used for inference-time self-reflection, supervised fine-tuning (SFT), and direct preference optimization (DPO).

**Strengths:**

This paper tries to address the problem of verifying the step-by-step logical integrity of CoT, not just the final answer. The mechanism for generating implicit premises to fill logical gaps is multi-step beyond simple auto formalization. Verification, when it passes, is a high-precision signal, filtering incorrect reasoning and improving reliability over the base model. The authors demonstrate the utility for model improvement, using the verification signal for self-reflection and as a high-quality filter for SFT/DPO data. The error analysis (Ungrounded, Contradiction, Untranslatable..) is insightful and provides actionable feedback for the LLM self-reflection loop.

**Weaknesses:**

1. The system "verifies" an LLM-generated CoT against LLM-generated premises. This is an internal consistency check, not a sound grounding check. There is lots of literature about LLMs preferring their own outputs.  A human evaluation or ground truth is required to measure the error rate of the autoformalizer and the premise generator to prove the system isn't just "consistently wrong."
2. The low pass rate on LegalBench-SARA is concerning. Qualitative error analysis is needed to explain this failure.
3. Writing could be improved for clarity. For example, Score in Tab 2 is not defined anywhere in the main text of the paper. Formatting errors are present, such as \approx in Line 425.

**Questions:**

To me, the core drawback to be fixed is that the paper must include a direct comparison against other neurosymbolic SMT-based CoT verifiers cited in the related work, namely SAT-LM (Ye et al., 2023), Logic-LM (Pan et al., 2023), Proof of Thought (Ganguly et al., 2024; 2025) and Logic-LM++ (Kirtania et al., 2024). On cursory browsing, current leaderboards for ProofWriter with Proof of Thought claim higher verified performance with GPT-5 than reported in your paper, and the SMT-LIB generation traces seem very similar. Without comparing against these techniques, the novelty and performance of VERICOT are unverified.

---

> ### Author Response · Authors · 2025-11-21
>
> We sincerely thank Reviewer C6Nf for the questions and comments. Here we address your concerns.
>
> **Q1. Need for direct benchmarking against existing neurosymbolic SMT-Based CoT verifiers**
>
> Thank you for pointing out these relevant SMT-based works. We have added the references to the draft, drawn comparisons, and updated the related work section accordingly. We believe there is a misunderstanding regarding the goal of our paper. The reviewer may be assuming that VeriCoT is intended to be a new neurosymbolic SMT-based reasoning system whose goal is to improve final task accuracy. In contrast, the purpose of VeriCoT is fundamentally different: VeriCoT does not aim to solve tasks or outperform prior SMT-based solvers. Its goal is to evaluate the validity of an existing CoT, regardless of the model that produced it, in a way that uses symbolic tools to extract the CoT’s core logic and underlying premises.. This goal is most similar to Explanation Refiner (Quan et al., https://aclanthology.org/2024.emnlp-main.172.pdf – EMNLP Outstanding Paper).
>
> The prior works mentioned by the reviewer (SAT-LM, Logic-LM, Proof-of-Thought, Logic-LM++) are methods that let LLMs generate solver code during reasoning with a focus on task success, i.e., generating reasoning traces with verified components that maximize benchmark performance. Those systems generate their own symbolic traces and report accuracy on the corresponding final answers. VeriCoT, however, takes a NL CoT as input and determines whether that reasoning is logically valid. This distinction makes direct leaderboard comparison inappropriate: the objectives, inputs, and outputs of VeriCoT differ from those of neurosymbolic reasoning systems.
>
> To evaluate CoT verification, rather than task-solving, we follow prior verification work (Explanation Refiner) in reporting pass rate (how often CoTs are deemed verifiable) and additionally introduce precision (proportion of correct answers among verifiable CoTs) and VCAR (pass rate × precision) to better capture verification quality to the extent that ground-truth answer labels are correlated with correct reasoning.
>
> **W1. LLM-only consistency checks cannot ensure soundness without human or ground-truth validation**
>
> We agree with the reviewer that asking an LLM to judge premises, especially ones still produced by an LLM, raises questions about consistency and bias. However, large-scale evaluation in practice requires automated LLM checks. Our approach mitigates this by only requiring LLM judgments on simple, atomic tasks. The SMT solver handles all formal reasoning, while the LLM is only asked to perform simple translation under a well-designed algorithm, generate premises from well-defined context, and perform smell checks on isolated premises. Prior work (e.g., FactScore, https://aclanthology.org/2023.emnlp-main.741.pdf ) shows that LLMs are much more reliable on such atomic judgments than on evaluating long chains of reasoning. This decomposition avoids compounding errors and also surfaces clear, human-readable premises, enabling targeted and transparent human review when needed.
>
> We conducted a human analysis by sampling 20 examples per dataset, evenly across the four labels: verifiable, ungrounded, untranslatable, and contradiction. We assess whether each logic translation is accurate, faithfully captures all necessary details, and preserves the correct logical relationships. We also evaluate the quality of the generated premises: (1) whether a contextual premise is correctly grounded in the source document or question, and (2) whether a commonsense premise is acceptable and reasonable given the context and the targeted reasoning step. Finally, we check whether LLMaaj’s judgments align with human annotations. Across all criteria, we observe consistently high performance. The autoformalizer exhibits a relatively low error rate, under 20%, which is already quite strong for NL-to-FoL translation. LLMaaJ decisions on premises are aligned with human judgements 95-100% of the time. We have updated the draft to reflect this.
> | Dataset      | Accurate translation / all statements | Grounded contextual premises / all premises | Acceptable commonsense premises / all premises | Correct LLMaaJ / all premises |
> |--------------|----------------------------------------|--------------------------------------------|-----------------------------------------------|--------------------------------|
> | BioASQ       | 0.80                                   | 0.87                                       | 0.91                                          | 0.97                           |
> | SARA         | 0.86                                   | 0.92                                       | 0.89                                          | 0.95                           |
> | Proofwriter  | 0.87                                   | 0.95                                       | 0.95                                          | 1.00                           |

---

> > ### Author Response · Authors · 2025-11-21
> >
> > **W2. Low pass rate on LegalBench-SARA**
> >
> > We conducted a qualitative analysis and found that one major issue comes from the SARA numeric subset, where the CoT chains are extremely long, exceeding 10 steps. In addition, even slight numerical differences (e.g., small differences in decimal places, particularly when rounding is involved) can cause the checks to fail. Here is one example:
> >
> > ```
> > Question: How much tax does Alice have to pay in 2017?. State the amount first.
> >
> > Document:  Statute: §3301. Rate of tax. There is hereby imposed on every employer (as defined in section 3306(a))...
> > (5000 words omitted)..
> >
> > Description: In 2017, Alice was paid $117192. Alice and Bob got married on Feb 3rd, 2017. Alice was a nonresident alien from August 23rd, 2016 to September 15th, 2018. Bob earned $37820 in 2017. Alice takes the standard deduction.
> >
> > CoT:
> > Step 1: Alice and Bob got married in 2017, but Alice was a nonresident alien during part of 2017
> > Step 2: According to tax law, a taxpayer is considered not married if their spouse is a nonresident alien during the tax year
> > Step 3: Therefore, Alice must file as single for tax purposes
> > Step 4: Nonresident aliens cannot take the standard deduction according to tax law
> > Step 5: Alice's gross income is $117,192
> > Step 6: For 2017-2025, the personal exemption amount is zero
> > Step 7: Alice's taxable income is $117,192 (gross income minus zero standard deduction and zero personal exemption)
> > Step 8: For single filers with taxable income over $115,000 but not over $250,000, the tax is $31,172 plus 36% of the excess over $115,000
> > Step 9: The excess amount over $115,000 is $2,192 ($117,192 - $115,000)
> > Step 10: The tax on the excess amount is $789 ($2,192 × 0.36)
> > Step 11: The total tax is $31,961 ($31,172 + $789)
> >
> > ```
> > In step 10, the value is automatically rounded to the nearest integer without explicitly stating that it is being rounded, so VeriCoT labels this step as ungrounded.
> >
> > **W3. Undefined scores in Tab 2**
> >
> > Thank you for the constructive feedback. We have updated the draft to define those scores to address your concerns. For the three scores reported in Table 2, as noted in Lines 358–359, we present the results of VeriCoT’s premise evaluation described in §2.4 (Lines 256–260). We use LLMaaJ to assess the quality of the generated premises: (1) grounded contextual premise: whether a contextual premise is correctly grounded in the source document or question, and (2) acceptable commonsense premise: whether a commonsense premise is reasonable given the context and the targeted reasoning step. In addition, we evaluate the necessity of each commonsense premise (necessary commonsense premise), i.e., whether it is required to support the corresponding reasoning step (Lines 361–364). We also added examples of different NL premises in Appendix A.3.

---

### Author Response · Authors · 2025-12-03

Dear Area Chair,

Thank you for taking the time to read our submission and the subsequent reviews.

**Summary**

Our paper introduces VeriCoT, **the first neuro-symbolic validator of CoT traces in non-math/code domains to our knowledge**. We also show how VeriCoT’s signals, used via self-reflection, SFT, and DPO, can improve the reliability of CoT reasoning.

All reviewers agree that **the paper offers a good contribution and is well presented**. Specifically, reviewer 37xu commented that _the paper’s motivation, proposed solution, structure, and writing quality are excellent_. Reviewer MriA commented that _the paper makes a nice coherent, and well-scoped contribution to test the reliability of LLM's CoT steps_.  Reviewer D7eV commented that _this concept demonstrates practical versatility, and suggests a path toward increasingly reliable reasoning systems_. Reviewer C6Nf commented that _the authors demonstrate the utility for model improvement and the error analysis is insightful_.

**List of revisions**

 In light of reviewer feedback, our main revisions are:

**1. An additional human analysis of LLM performance**

- **Reviewer concern:** Most reviewers suggested the need for additional human analysis of LLM performance from different perspectives, specifically regarding the error rates of the autoformalizer, premise generation, and premise evaluation.


- **Revisions to the paper:** We have added additional human analysis to the draft; the details can be found in Section 3.4 and Table 3. The autoformalizer exhibits a relatively low error rate, which is already quite strong for NL-to-FoL translation, and LLMaaJ decisions on premises are aligned with human judgements 95-100%.

**2. A more detailed related work section to disambiguate our work from previous works with different goals**

- **Reviewer concern:** Reviewer **C6Nf** compared VeriCoT to prior works whose goals we believe are not related. The reviewer may assume that VeriCoT is a new neuro-symbolic SMT-based reasoning system designed to improve task accuracy, i.e., generating reasoning traces with verified components while maximizing benchmark performance (similar to SAT-LM, Logic-LM, Proof-of-Thought, Logic-LM++ mentioned by the reviewer).

- **Revisions to the paper:**  We have updated Section 4 related work to make it clear that VeriCoT has a different purpose: it is not intended to solve tasks or outperform prior SMT-based solvers, but to assess the validity of an existing CoT, regardless of the model that produced it, by using symbolic tools to extract its core logic and underlying premises. This goal is most similar to Explanation Refiner (Quan et al., https://aclanthology.org/2024.emnlp-main.172.pdf  – EMNLP Outstanding Paper).

**3. A clearer explanation of the evaluation metrics**

- **Reviewer concern:** Reviewer **37xu** was concerned about our task metrics and advocated that precision is the most important metric and a high pass rate doesn’t necessarily mean better verification.

- **Revisions to the paper:** We have added additional justification to the evaluation metrics in Section 3.4. To evaluate CoT verification, rather than task-solving, we follow prior verification work (Explanation Refiner), in reporting pass rate. We additionally introduce precision and VCAR (pass rate × precision) to better capture verification quality. While the reviewer prioritizes precision, it must be understood within the precision–recall trade-off: increasing pass rate (recall) typically lowers precision. On ProofWriter, baselines achieve very high precision only by “cherry-picking’’ trivial cases and abstaining on most others, resulting in extremely low pass rates that undermine practical utility. VeriCoT, however, achieves a much higher pass rate while also maintains a high precision because it can handle more challenging questions. We also agree that "a high pass rate doesn’t necessarily mean better verification." Hence we propose VCAR, which captures both leniency and accuracy: it penalizes verifiers that are overly lenient (high pass rate, low precision) and those that are overly selective and accept only trivial cases (low pass rate, high precision).

We sincerely hope this reviewer-level summary will assist in your decision.

---

### Meta-Review · Area_Chair_bVBz · 2025-12-27

**Summary:**

This paper presents the first neuro-symbolic validator of CoT traces in non-mathematical or code domains, to our knowledge. I agree with reviewers that the overall presentation is clear and the contribution is solid. The primary concern reviewers have is that "The system "verifies" an LLM-generated CoT against LLM-generated premises." This concern is valid, but the authors have provided additional empirical evidence to support the effectiveness of the proposed method. Hence, I would recommend accepting this paper.

**Reviewer Concerns:**

1. The core contribution of this paper is not theoretically guaranteed (The system "verifies" an LLM-generated CoT against LLM-generated premises.)

**Reviewer Scores:**

I think reviewer 37xu might increase the score from 4 to 6. Others should remain the same.

---

### Decision · Program_Chairs · 2026-01-26

Accept (Poster)